# Impacts of marine heatwaves on top predator distributions are variable but predictable

Heather Welch [1,2] ✉, Matthew S. Savoca [3], Stephanie Brodie [1,2], Michael G. Jacox [1,2,4], Barbara A. Muhling[2,5], Thomas A. Clay[1,2,6], Megan A. Cimino [1,2], Scott R. Benson [7,8], Barbara A. Block[3], Melinda G. Conners[9], Daniel P. Costa[2,10], Fredrick D. Jordan [9], Andrew W. Leising[1], Chloe S. Mikles[3], Daniel M. Palacios [11,12], Scott A. Shaffer [13], Lesley H. Thorne [9], Jordan T. Watson[14,15], Rachel R. Holser [2], Lynn Dewitt [1], Steven J. Bograd [1,2] & Elliott L. Hazen [1,2,3]

Marine heatwaves cause widespread environmental, biological, and socio-economic impacts, placing them at the forefront of 21st-century management challenges. However, heatwaves vary in intensity and evolution, and a paucity of information on how this variability impacts marine species limits our ability to proactively manage for these extreme events. Here, we model the effects of four recent heatwaves (2014, 2015, 2019, 2020) in the Northeastern Pacific on the distributions of 14 top predator species of ecological, cultural, and commercial importance. Predicted responses were highly variable across species and heatwaves, ranging from near total loss of habitat to a two-fold increase. Heatwaves rapidly altered political bio-geographies, with up to 10% of predicted habitat across all species shifting jurisdictions during individual heatwaves. The variability in predicted responses across species and heatwaves portends the need for novel management solutions that can rapidly respond to extreme climate events. As proof-of-concept, we developed an operational dynamic ocean management tool that predicts predator distributions and responses to extreme conditions in near real-time.

Long-term climate trends (e.g., global warming) and short-term extreme events (e.g., heatwaves) have global impacts on ecosystem structure and functioning, and human well-being[1–3]. The impacts of long-term climate trends have received considerable attention through the examination of the warming signal in both historical observations and future climate projections[4–9]. However, mounting evidence indicates that episodic events like fires, floods, and heatwaves can have catastrophic ecosystem and socio-economic impacts[1,10–12].

[1]NOAA, Southwest Fisheries Science Center, Environmental Research Division, Monterey, CA, USA. [2]Institute of Marine Science, UC Santa Cruz, Santa Cruz, CA, USA. [3]Hopkins Marine Station, Stanford University, Pacific Grove, CA, USA. [4]NOAA, Physical Sciences Laboratory, Boulder, CO, USA. [5]NOAA Southwest Fisheries Science Center, Fisheries Resources Division, San Diego, CA, USA. [6]People and Nature, Environmental Defense Fund, Monterey, CA, USA. [7]NOAA, Southwest Fisheries Science Center, Marine Mammal and Turtle Division, Moss Landing, CA, USA. [8]Moss Landing Marine Laboratories, San Jose State University, Moss Landing, CA, USA. [9]School of Marine and Atmospheric Sciences, Stony Brook University, Stony Brook, NY, USA. [10]Department of Ecology and Evolutionary Biology, UC Santa Cruz, Santa Cruz, CA, USA. [11]Marine Mammal Institute, Oregon State University, Newport, OR, USA. [12]Department of Fisheries, Wildlife, and Conservation Sciences, Oregon State University, Newport, OR, USA. [13]Department of Biological Sciences, San Jose State University, San Jose, CA, USA. [14]NOAA, Alaska Fisheries Science Center, Auke Bay Laboratory, Juneau, AK, USA. [15]Pacific Islands Ocean Observing System, University of Hawai'i Mānoa, Honolulu, HI, USA. ✉e-mail: heather.welch@noaa.gov

In particular, heatwaves on land[13] and at sea[1] adversely impact an additional 157 million people today compared to the turn of the century[14]. The largest and warmest marine heatwaves (MHWs) on record have occurred in the last decade and include the 2012 Northwest Atlantic[15], the 2015–16 Tasman Sea[16], and 2013–16 Northeast Pacific[17] events, exerting considerable physical, ecological, socio-economic, and human health impacts[1].

Faced with a changing climate, mobile species' first responses are often to shift their geographic ranges to remain within suitable environmental conditions[2]. This phenomenon is well-described within the context of long-term warming[4–6,8] and El Niño events[18,19], yet there is a paucity of information on species redistribution in response to unprecedented warming during recent MHWs[20]. Temperatures observed during MHWs can be similar to projected mean future conditions at the end of the 21st century[21], providing valuable insight into species redistribution in the coming decades. Previous investigations on MHW-driven species range shifts have largely relied on observational data from opportunistic sightings, surveys, or tagging programs, and have demonstrated poleward or vertical distributional shifts towards cooler temperatures[22–26]. However, these data are often patchy, offering only snapshots of impacts from a single MHW on a single species[22,23,27] or on several species[24,25,28]. Comparisons of multi-species responses across multiple MHWs are rare despite ample evidence that there is high variability in the evolution, drivers, and physical characteristics of MHW events[21,29,30]. Statistical models provide a means of interpolating across space, time, and taxa, providing information on MHW-driven redistribution by offering inferences on unobserved locations, MHWs, and individuals. Furthermore, statistical models can relate species distributions to multiple environmental drivers, thereby accounting for the complex physical and biogeochemical changes beyond the increased temperature that occurs during MHWs[29].

Here, we quantify the impacts of four major North Pacific MHWs (2014, 2015, 2019, 2020; Supplementary Fig. 1) on the spatial distributions of 14 marine top predators, spanning several major guilds: seabirds, mammals, turtles, tunas, and sharks (Fig. 1). The impacts to most of these species during the MHWs were previously unexamined,

and thus, we provide an unprecedented look at how responses vary across a group of iconic species and across MHWs.

The Northeast Pacific Ocean provides an ideal testbed for examining the effects of MHWs. This region has experienced some of the longest, largest, and most intense MHWs on historical record[31,32] (Fig. 1A), with thermal displacements of >1000 km during the strongest events[21]. The Northeast Pacific is also a biodiversity hotspot and major foraging ground[33], attracting a range of top predator species from across the broader Pacific Ocean, many of which are ecologically and commercially valuable and/or threatened with extinction[33] (Fig. 1B). These predators are highly mobile and conduct large-scale movements throughout the Pacific basin[34], and thus have the physical ability to actively redistribute in response to MHWs. Many of the most extreme temperature anomalies have occurred where predator density is highest (Fig. 1), providing an opportunity to examine MHW impacts across several higher trophic levels that are particularly sensitive to climate perturbations[35]. Furthermore, we examine how MHWs shift predators across political boundaries, revealing which exclusive economic zones (EEZs) lost and gained predator habitat and where adaptable governance strategies are likely to be important. By implementing a multi-species, multi-MHW analytical framework, our work provides the basis for informing future management of the impacts of these extreme climatic events.

## Results

We used boosted regression tree models fit extensive telemetry datasets[33,36,37] to predict the redistribution of species' preferred habitats during each MHW event. Our models accurately predicted distribution shifts during MHW years captured by an extensive (>one million records) independent top predator dataset collated from public, private, and government sources (Supplementary Table 2 & Supplementary Fig. 7). This independent dataset included records from tagging programs, shipboard surveys, opportunistic sightings, and fisheries observer programs, and allowed us to explore predictive performance in the telemetry-based models. For some species, independent datasets were more closely aligned with those used to build

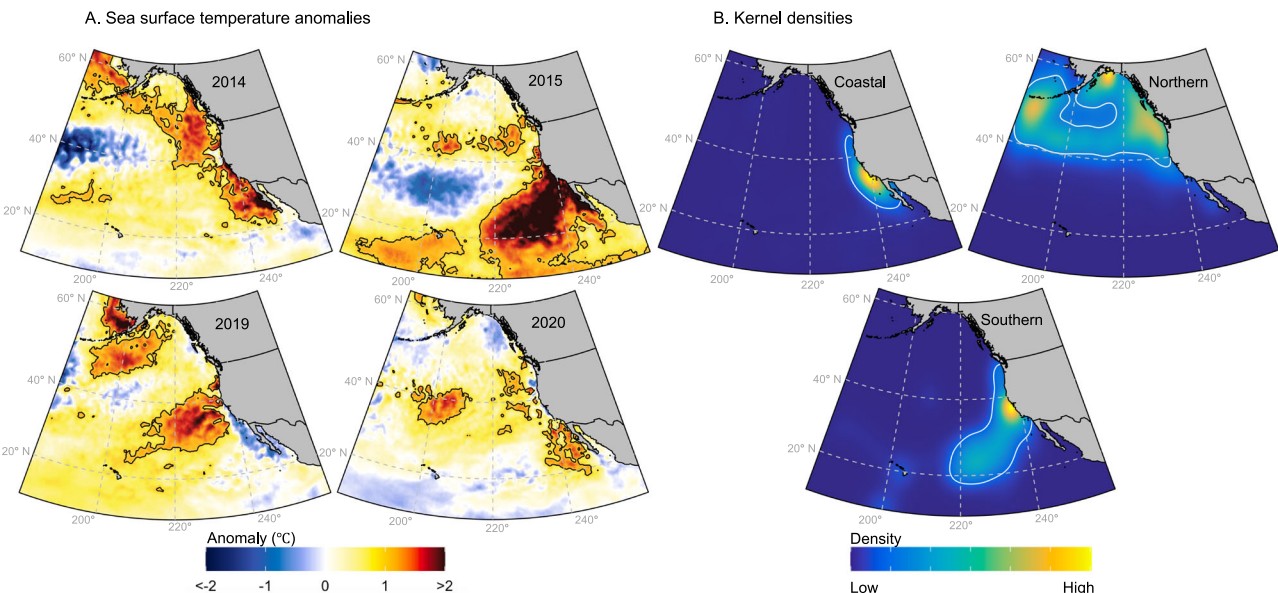

**Fig. 1 | Marine heatwave (MHW) and top predator distributions in the North Pacific. A** Mean sea surface temperature anomaly in August–October (the months in which the highest temperature anomalies were observed across the Pacific), calculated relative to a 2000–2020 baseline for each of the four MHW events explored, with 1.5 °C contour in black. **B** Species kernel densities while foraging and transiting in the Northeastern Pacific, grouped by the location of animal tracking data (2000–2010). The 75th percentile kernel for each species group is shown by the white contour. Coastal species include blue and mako sharks, yellowfin, albacore, and bluefin tuna, California sea lions, sooty shearwaters, and blue whales; Northern species include elephant seals, salmon sharks, black-footed and Laysan albatross; Southern species include white sharks and leatherback turtles.

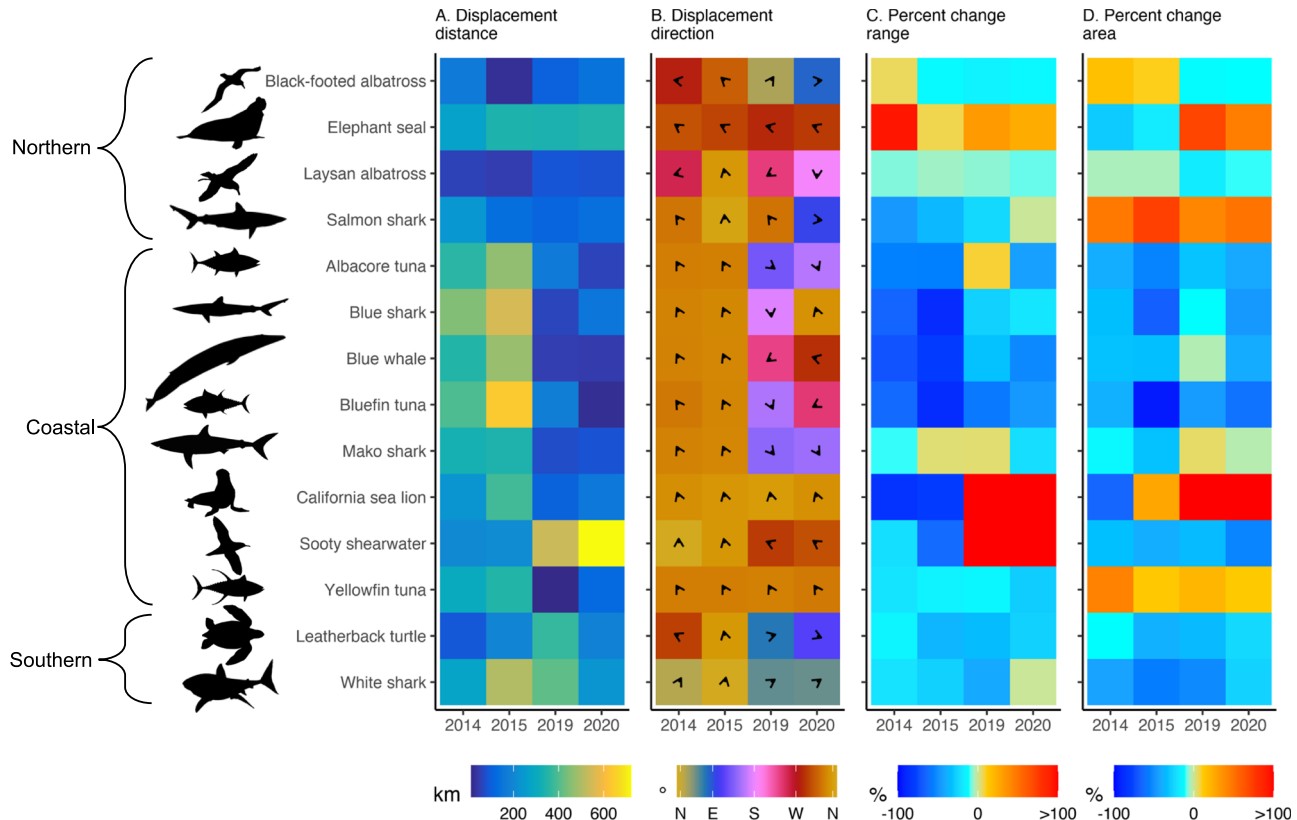

**Fig. 2 | Predicted impacts on top predator habitat within (columns, e.g., 2014) and among (rows, e.g., White shark) marine heatwave events measured using four metrics. A** Displacement distance (kilometers), **B** displacement direction (degrees, where 0/360 is north (N), 90 is east (E), 180 is south (S), and 270 is west (W)), **C** range compression or expansion (percent change relative to baseline conditions), **D** habitat area gain or loss (percent change relative to baseline conditions). All metrics were calculated from August–October in each MHW year relative to baseline conditions (August–October 2000–2020), see Supplementary Table 5 for an analysis of metric uncertainty. Northern, Coastal, and Southern regional groupings indicate the geographies where the majority of the species telemetry data occurs. Source data are provided as a Source Data file.

models due to a similar spatial domain and/or data type (e.g., telemetry data for blue whales), which resulted in higher predictive performance. However, for others, opportunistic sightings and shipboard surveys covered different spatial domains than the telemetry datasets and typically had lower predictive performance (e.g., Laysan albatross, Supplementary Fig. 8).

Predicted MHW impacts were highly variable within and among MHW events (Fig. 2). For example, responses among MHWs (rows, Fig. 2) were highly variable for Coastal species. While all Coastal species were displaced to the northwest during the 2014 and 2015 events, the 2019 and/or 2020 events drove southeastward displacement for bluefin and albacore tuna, and blue and mako sharks (Figs. 2B and Fig. 3C; Supplementary Fig. 10A, C, I). Southeastward displacement may be related to the emergent cool water refugia along the southern US and Mexican coasts during these two events (Fig. 1A). Sooty shearwaters and sea lions underwent a reversal in range shifts between the 2014–15 and 2019–20 events, experiencing range compression during the 2014 and 2015 events (22–64% decrease), and range expansion during the 2019 and 2020 events (335–377% increase) (Figs. 2C and 3A; Supplementary Fig. 10E). Similarly, elephant seals and sea lions lost habitat during the 2014 MHW (29% and 65% decrease, respectively), but gained habitat during the 2019 and 2020 MHWs (71–46% and 117–158% increase) (Fig. 3B; Supplementary Fig. 10E).

In addition to among-MHW differences, we found differences in predicted impacts within-MHWs (columns, Fig. 2). While most species were predicted to experience relatively low displacement distances during the 2019 and 2020 events, sooty shearwaters were displaced large distances (536 and 721 km, respectively) during these two MHWs. Displacement direction during the 2020 event ranged from north

(blue shark), east (salmon shark), south (Laysan albatross), and west (blue whale) (Fig. 2B; Supplementary Fig. 10). The 2019 and 2020 events caused large range expansions for sooty shearwaters and California sea lions whereas most other species experienced range compression (Figs. 2C and 3A; Supplementary Fig. 10E). While most species lost habitat in 2019 and 2020, elephant seals, salmon sharks, California sea lions, and yellowfin tuna gained habitat (Fig. 2D).

Among- and within-MHW differences were also apparent in the predicted redistribution of predators across political boundaries (Fig. 4). The US EEZ was predicted to gain predator habitat during each MHW, with the largest increase during the 2015 event (10%), and the smallest increase during the 2020 event (2%; Fig. 4A). The Mexican EEZ was predicted to lose predator habitat during 2014, 2015, and 2020 events (with the largest decrease in 2015 of 8%). The Canadian EEZ and high seas were predicted to gain and lose predator habitat during each MHW, respectively. Coastal species experienced the largest jurisdictional redistributions during the 2014 and 2015 events, particularly yellowfin and albacore tunas, with 31% and 22% of their respective habitats predicted to shift into US waters (Fig. 4B). Southern species experienced the largest redistributions during the 2019 event, with 39% of white shark habitat and 8% of leatherback turtle habitat predicted to shift into the US EEZ. Northern species experienced the largest redistributions during the 2020 event, with 14% of salmon shark habitat predicted to shift into the Canadian EEZ.

## Discussion
We used a standardized framework to quantify the impacts of multiple MHWs on a numerous and diverse set of top predators. This is an important step because MHWs vary in intensity and evolution, and

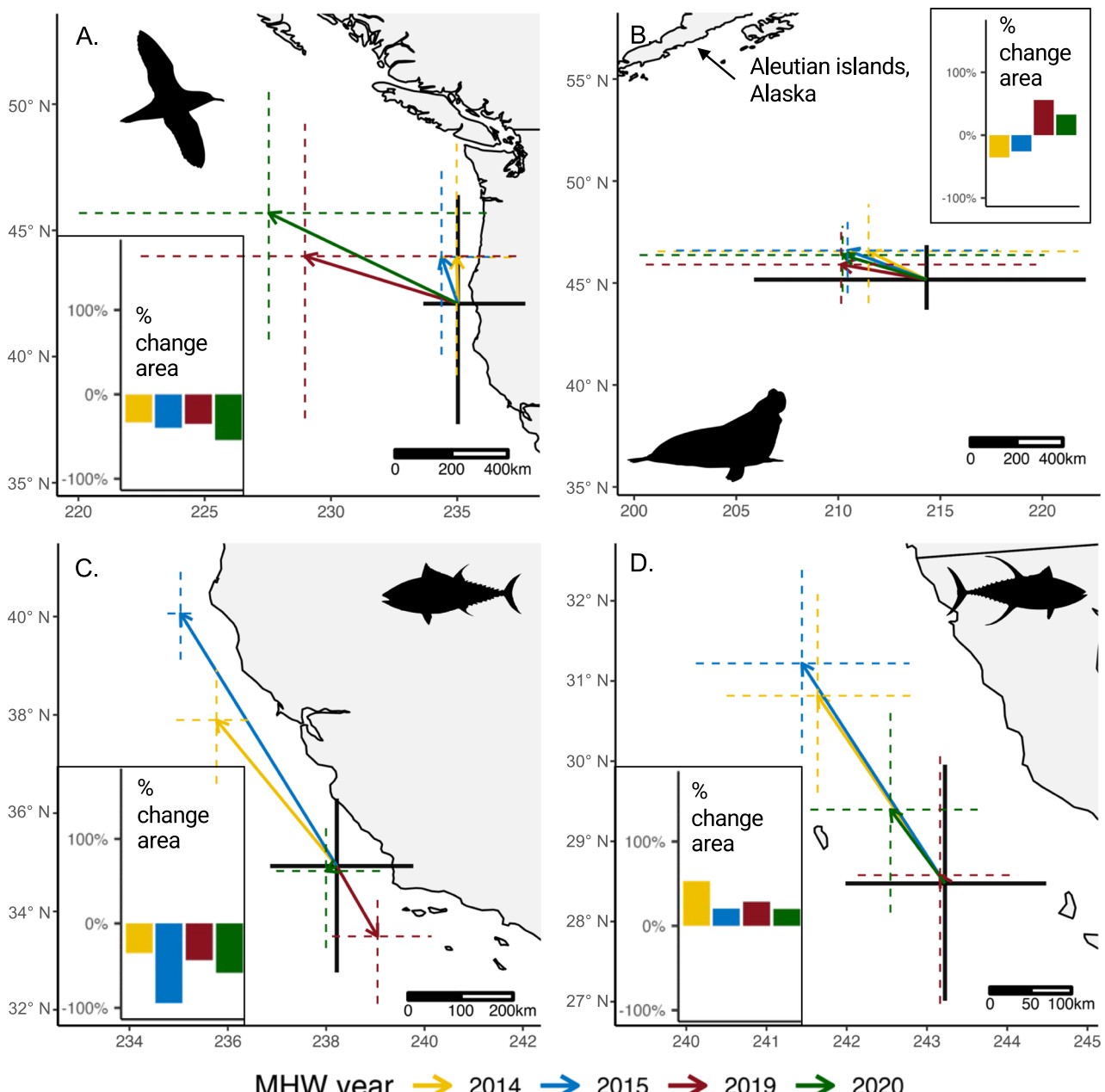

**Fig. 3 | Marine heatwave (MHW) impacts on four species. A** Sooty shearwaters, **B** Elephant Seal, **C** Bluefin tuna, and **D** Yellowfin tuna. Large maps: arrows indicate predicted habitat displacement from the center of gravity in each MHW (center of colored crosses) relative to the center of gravity during baseline conditions (center of black cross). Crosses indicate predicted longitudinal and latitudinal range extents during each MHW and during baseline conditions (colored and black crosses, respectively); an increase in cross size during a MHW compared to baseline conditions indicates range expansion with the converse representing compression. Inset: percent change habitat area relative to baseline conditions.

predators vary in their relationship to the environment. While previous studies have examined species redistribution in response to past climate variability[18,19], few have done so in a standardized way across a large range of predator guilds. Here, predicted responses were highly variable across species and MHWs (Fig. 2): some predators were predicted to experience near-total loss of habitat and range compression, e.g., bluefin tuna during the 2015 event, while others were predicted to experience a two-fold habitat increase and significant range expansion, e.g., California sea lion during the 2019 event. Critically, this MHW-induced habitat loss exceeds the projected habitat loss by the year 2100 for these same species due to long-term warming[4], though we note that there are differences in methodologies between these studies.

The most severe species impacts tended to occur in regions where MHWs temperature anomalies were highest (Figs. 1 and 2; Supplementary Fig. 3). Coastal and Southern species were predicted to experience particularly warm temperatures during the 2014 and 2015, and 2015 and 2019 events, respectively, leading to large displacement, range compression, habitat loss, and cross-jurisdictional shifts. Northern species were predicted to experience more moderate temperature anomalies and habitat impacts relative to Coastal and Southern species. However, temperature anomalies were fairly consistent across the 75th percentile kernels of sooty shearwaters and elephant seals during the four MHW events, yet each species experienced markedly different predicted impacts among MHWs. Sooty shearwaters were predicted to expand their range during the 2019 and

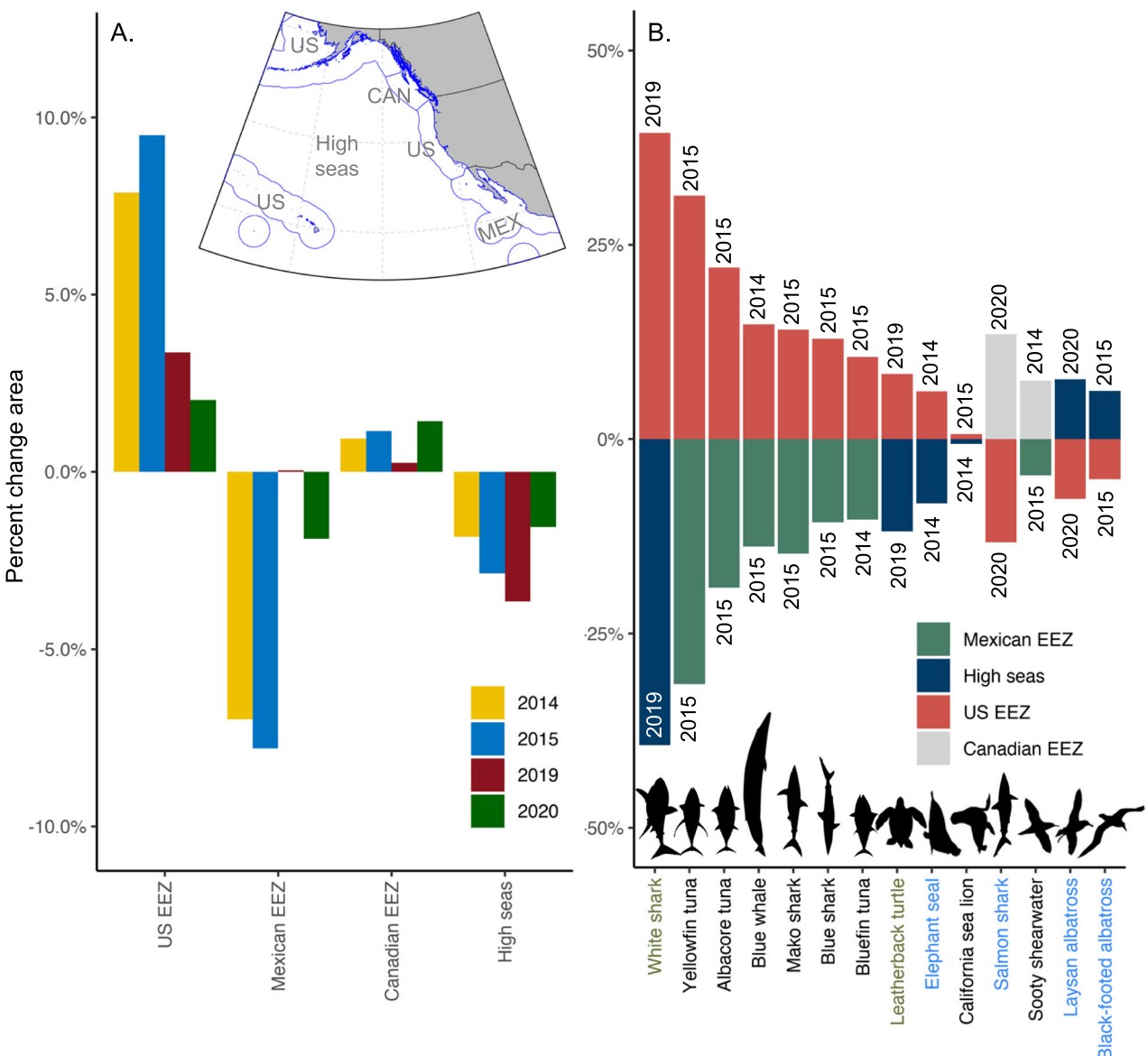

**Fig. 4 | Cross-jurisdictional shifts in predators' predicted habitats during marine heatwaves (MHWs). A** Total loss and gain of species habitat area across the US, Mexican (MEX), and Canadian (CAN) exclusive economic zones (EEZs), and the high seas during each MHW. **B** Largest loss and gain of habitat for each predator in any MHW event (*x*-axis text color indicates species regional groupings: Southern (green), Coastal (black), Northern (blue)). Percent change in habitat area is calculated relative to baseline conditions (2000–2020), see Supplementary Table 7 for an analysis of uncertainty in cross-jurisdictional shifts. Source data are provided as a Source Data file.

2020 events, likely associated with elevated primary productivity in coastal Alaskan waters (Supplementary Fig. 2). Elephant seals were predicted to experience large habitat loss during the 2014 and 2015 events, likely due to a reduction of primary productivity and subsurface oxygen in the north-central Pacific (Supplementary Fig. 2). The deep warm water anomalies of the 2014 and 2015 events decreased the solubility of oxygen and led to a near-surface decline in oxygen concentration[38]. The 2019 and 2020 events, in contrast, had much shallower temperature anomalies and likely had a weaker impact on local biogeochemical signals and productivity. In general, variations in winds, air-sea gas exchange, circulation patterns, and water column stratification can lead to very different physical and biogeochemical conditions among MHW events. The idiosyncratic responses of species to MHWs reflect these variable conditions and differing species-environment relationships (Supplementary Fig. 5).

MHW-driven species redistributions are most commonly ascribed to the relocation of preferred temperature conditions[23,26,39]. However,

our findings suggest that a multi-variable approach allows for additional ecological inferences with regard to MHW biodiversity impacts. Many of the predators examined here have broad thermal tolerances and are distributed in warmer waters elsewhere in the Pacific than those encountered during the MHWs[33]. Thus it is likely that anomalous conditions of variables beyond temperature (Supplementary Fig. 3) are driving their responses. Indeed, a comparable suite of temperature-only models had a significantly worse predictive performance on novel validation data than the multivariate models (median Area Under the Receiver Operator Characteristic Curve of 0.6 in the temperature-only models vs 0.8 in the multivariate models, t-test p-value < 0.05). In following with evidence that temperature alone cannot account for species responses to climate change[6,40], we suggest that a multi-variable approach is critical to capture species responses to short-term warming. Several programs exist to monitor and forecast extreme ocean warming based on observed and predicted ocean temperatures[41,42], and these results indicate the utility of concurrently

tracking ocean conditions such as oxygen and productivity for a more nuanced understanding of possible species impacts.

## Inferences from models

Correlative species distribution models such as those used in the present study are valuable tools to understand species ecology and are often used to support environmental conservation and management across multiple time horizons[4–6,33,43,44]. Importantly for MHWs, models allow for insights into the complex environmental relationships that drive species responses. However, the correlative models used here do not explicitly capture species traits such as physiology, movement syndromes, and life-histories[45]. For example, ocean wanderers like sharks, tunas, elephant seals, and albatrosses may be buffered from MHW effects due to their generalist diets and ability to exploit distant waters[26,46]. The effects of MHWs on central place foragers that are tied to a colony during the breeding season (e.g., sea lions) may be greater, as these species are less capable of accessing distant patches of suitable habitat. The models captured species fundamental niches, the full set of environmental conditions where a species foraged or transited within the North Pacific. However, they did not account for their environmental preferences during reproductive behaviors (e.g., some species migrate to the West Pacific for spawning or nesting) or other ecological processes that may affect distribution (e.g., prey availability, interspecific interactions, population structure, or site fidelity)[45].

## Transboundary predators

Long-term warming has redistributed species across EEZ boundaries, and is projected to continue to redistribute species in the future[5]. Here, we demonstrate that MHWs may drive jurisdictional shifts over more immediate time-frames, leading to new national risks, rewards, and responsibilities. During each event, our models predicted an influx of species habitat into the US EEZ from the Mexican EEZ and the high seas (Fig. 4A). Predators also shifted into the Canadian EEZ during each MHW, though to a lesser extent than that of the US. The magnitudes of predicted losses and gains of predator habitat varied by MHW event, indicating that transnational management will need to be dynamic and adaptable. Conflicts over shifting transboundary stocks have occurred elsewhere, for example, the northeast Atlantic mackerel (*Scomber scombrus*) wars[47], offering precautionary tales for the value of coordinated and proactive management across nations.

These findings indicate that the US in particular, will face new management challenges during MHWs. Between 10 and 31% of the predicted habitat of commercially valuable albacore, bluefin, and yellowfin tuna shifted from Mexico to the US (Fig. 4B); indeed, an unusual abundance of yellowfin and bluefin tuna was reported by California commercial and recreational fishers during the 2014 and 2015 event[25]. These episodic fishing opportunities will require rapid managerial oversight to ensure stock sustainability, e.g., a climate-driven tilefish redistribution along the US east coast led to the stock being exploited without regulation for nearly a decade[5]. Predicted core habitats of protected species also shifted into US waters: elephant seal and leatherback turtle habitats were redistributed to the US EEZ from the high seas. Both species are bycaught in US fisheries (although bycatch rates are currently low), and may require increased monitoring to ensure fisheries interactions do not threaten their populations. Protected blue whale habitat shifted from Mexican to US waters, potentially increasing management concern over mortality from ship-strikes and entanglement in fishing gear which has been observed in US waters during MHW conditions[48,49]. Up to 39% of threatened white shark habitat shifted into the US waters from the high seas—the largest redistribution of any species (Fig. 4B). Although redistributed white sharks may benefit from low bycatch rates in US waters, higher white shark prevalence may lead to increasing predation rates of protected pinnipeds and associated ecosystem effects such as reduced kelp cover[50].

## Early warning systems

Our results show a wide range of species impacts across MHW events and jurisdictions. This diversity of responses poses a daunting management challenge: how to plan ahead and respond swiftly to MHW-driven species redistribution. Our results indicate that species responses to MHWs are highly variable yet predictable: our models performed well through extensive validation across space, time, and on novel data (Supplementary Figs. 6 and 7; Supplementary Table 3). The high variability of species responses to MHWs suggests that we cannot assume future MHWs will impact species in the same way as past events. However, high predictability indicates that species responses to future MHWs could be predicted in real-time to provide accurate information on impacts. This combination of high variability and predictability is also seen in hurricanes—future hurricane paths cannot be extrapolated from past paths, yet they can be predicted in real-time to provide accurate information on risk.

Dynamic ocean management[51] tools are designed to translate changing environmental and biological information into real-time management recommendations, and have shown promise at keeping pace with anomalous ecological conditions during MHW events[39,43,44]. These tools frequently rely on near real-time predictions from species distribution models to capture current ecological conditions[52]. As a proof of concept, we have operationalized the top predator models to produce daily predictions of each species' current distribution: https://oceanview.pfeg.noaa.gov/top-predator-watch/. This operational framework could be rapidly integrated into a dynamic ocean management tool to address MHW-driven human-wildlife conflicts in real-time, similar to several tools already in applied use[43,44]. Successful monitoring of MHW impacts requires continued observation of species responses, which in the past has proved to be more opportunistic than by design. Real-time predictions of species distributions during MHWs could be used to guide observational programs (e.g., field surveys and tagging programs) during anomalous conditions. Importantly, these new observations could be assimilated with distribution model outputs to improve real-time model predictions, a common process in oceanography, which is not yet a standard practice in ecology[53].

Forecasts of future impacts would offer longer lead times for decision-making compared to real-time predictions. Extreme episodic events like hurricanes, fires, and floods are successfully forecast on land[54,55], and skillful forecasts of MHWs have recently become operational at lead times of up to a year, depending on the region[41,56]. The next frontier is to skillfully forecast ecological variables, including species distributions, in response to these extreme episodic events. These early warning systems would allow for proactive—as opposed to reactive—responses to new human-wildlife conflicts, changing marine resource availability, and emergent refugia caused by MHWs[49,57], allowing us to plan ahead for our fundamentally dynamic world.

## Methods

Telemetry data used to build the species distribution models (SDMs) were acquired for 14 top predators tagged from 2000–2010, including data from the Tagging of Pacific Predators Project[33] and private datasets. For all species except albatrosses, detailed methods on the number of individuals tagged, handling of tagging bias, and state space modeling are included in Block et al.[33], Winship et al.[36], and Jordan et al.[37]. Further details on albatross telemetry data are available in Supplementary methods section 1.3.

Species were assigned to one of three geographical groups—Northern, Coastal, and Southern—based on the locations of telemetry data collected from August–October across all years. Coastal species had over 60% of observations located within the California Current and Gulf of California Large Marine Ecosystems (blue [*Prionace glauca*] and mako [*Isurus oxyrinchus*] sharks; albacore [*Thunnus alalunga*], yellowfin [*T. albacares*], and bluefin [*T. thynnus*] tunas; blue whales

[*Balaenoptera musculus*], sooty shearwaters [*Ardenna grisea*], and California sea lions [*Zalophus californianus*]), despite some of these species undertaking long migrations across the North Pacific. Northern species had observations to the northwest of Coastal species (black-footed [*Phoebastria nigripes]* and Laysan [*P. immutabilis]* albatross; elephant seals [*Mirounga angustirostris*] and salmon sharks [*Lamna ditropis*]); and Southern species had observations to the southwest (white sharks [*Carcharodon carcharias*] and leatherback turtles [*Dermochelys coriacea*]). These regional groupings were useful to describe broad patterns, as MHW impacts were most similar across species within the same geographical group. The 75th percentile kernels for each geographical group were calculated by randomly subsampling each species' locational data to the same number of records, and then calculating kernel densities and the 75th percentile kernel across all species within each grouping (Fig. 1B). Kernel densities and 75th percentile kernels were also calculated for each species individually and used to quantify environmental conditions within each species' 75th percentile kernel during the MHW events (Supplementary Fig. 3).

For each year, MHWs were assessed from August–October: the months in which the highest temperature anomalies were observed across the Pacific (Supplementary Fig. 1, Supplementary methods section 1.2). In addition to SST anomalies (Fig. 1A), anomalies of oxygen at 200 m, mean primary productivity within the upper 200 m, and surface chlorophyll-a conditions were also assessed during MHWs (Supplementary Table 1). Anomalies during each MHW were calculated relative to mean conditions across August–October 2000–2020. We were precluded from using a longer baseline by the availability of satellite observations for chlorophyll-a (a covariate in the SDMs), which came online as a science-quality product in 1998.

The boosted regression tree (BRT) models used the multi-year animal telemetry data to describe species-environment relationships from a suite of environmental variables (Supplementary Table 1, Supplementary Methods section 1.1). Dynamic variables used to fit the BRTs included primary productivity, oxygen, sea surface temperature (SST) and its spatial standard deviation, sea level anomaly, eddy kinetic energy, mixed layer depth, chlorophyll-a, and day of the year. All environmental variables were resampled from their native resolutions to 0.25 degrees to match the coarsest resolution of the environmental datasets (sea surface height products). All analyses were performed in R version 4.0.4.

Background pseudo-absences were generated at a 1:1 ratio of presences for the telemetry datasets[58]. For each presence point, a pseudo-absence was generated for the same date (Supplementary methods section 1.5.1). Presence and pseudo-absence data were matched to the environmental datasets in space and time. We used a BRT framework to model the probability of species presence as a function of the environment. BRTs are a common machine learning model, popularized by their ability to fit complex nonlinear relationships and their robustness to wide varieties of data types and distributions. For all 14 species, BRTs with a binomial distribution were used to model the probability of species presence as a function of the environment. BRTs were built with a bag fraction of 0.6, and a tree complexity of three, and a learning rate that varied between 0.0001 and 0.00001 to ensure at least 2,000 trees were fit for each model. In addition to predicting suitable habitats, the relative importance of variables in each SDM was identified (Supplementary Fig. 5).

Independent datasets were used to validate the temporal extrapolation of telemetry-derived models beyond the time-series of the training data (i.e., post-2010), particularly during MHW years (Supplementary Table 2, Supplementary methods sections 1.4. and 1.5.4). Datasets were acquired from a diverse range of public and government sources including fisheries observer programs, animal sightings from citizen science databases, and tagging data from dedicated programs. For each species, all data sources were pooled for validation, e.g., the blue shark model was validated on an independent dataset composed of both California drift gillnet observer data and survey data from the North Pacific Pelagic Seabird Database (Supplementary Table 2). In addition to validation on independent data, several methods of cross-validation were performed (Supplementary Table 3, Supplementary methods section 1.5.3).

The BRT models were predicted over the daily environmental data from 2000–2020, and spatially constrained within a minimum convex hull of the training data. The resultant habitat suitability predictions ranged from 0–1, with low and high values indicating unsuitable and suitable habitats, respectively. These continuous predictions were averaged to create an August–October 2000–2020 mean, and August–October anomalies during each MHW year were calculated relative to the long-term mean (Supplementary Fig. 9).

The continuous habitat suitability predictions were reclassified into daily binary core habitat using species-specific thresholds (see Supplementary methods section 1.6.2 for a comparison of binary versus continuous core habitat). Core habitat was defined as pixels with habitat suitability greater than or equal to the top 50% of predicted values at true presences (i.e., models were predicted back on the telemetry data used in model fitting, and 50% prediction quantile was used as the threshold). The 50% threshold was selected over the more conservative thresholds used in the climate projection literature (e.g., 25%[4]) as these conservative thresholds sometimes resulted in a complete loss of core habitat during extreme events (e.g., 2015).

We quantified MHW impacts on species' core habitats using four metrics (Fig. 2). Displacement direction and distance captured how the center of gravity of core habitat changed during MHWs, and was measured as a vector of both cardinal direction (degrees) and distance (kilometers). Range extent captured the distance between the leading and trailing edges of the core habitat in the north-south and east-west directions, providing a metric of range compression or expansion due to MHWs (negative and positive percent change range, respectively). Core habitat area captured the total available core habitat and its loss or gain due to MHWs (negative and positive percent change area, respectively). These metrics are independent of one another; for example, during a MHW, a species can gain new habitat beyond its typical range (i.e., range expansion), while overall having less available habitat throughout its range (i.e., habitat loss).

Three metrics were calculated from the daily binary core habitat rasters: the center of gravity of core habitat (mean latitude/longitude coordinate pairs), the interquartile range of core habitat (latitude/longitude coordinate pairs for the north-south and east-west interquartile ranges), and the total amount of core habitat (km²). To quantify species' range extents, the daily north-south interquartile range was multiplied by the daily east-west interquartile range (both measured in decimal degrees). The interquartile range was selected over the absolute range to remove the effect of outliers (e.g., individual pixels of core habitat with extreme distributions), and to ensure that expansion of range was not artificially truncated due to proximity to the convex hull boundary.

The three metrics were averaged across each MHW event (August–October of 2014, 2015, 2019, 2020), and deviations from the 2000–2020 August–October mean was calculated for each habitat metric and each species, producing estimates of habitat displacement, changes to the range extent, and changes to the habitat area. Habitat displacement was assessed as a vector (i.e., cardinal direction and distance) joining the long-term mean position of the core habitat center of gravity to its MHW-specific location. Range extent change reflected the MHW-specific compression or expansion of range extent compared to the mean range, measured by area, while habitat area change assessed the MHW-specific loss or gain of total core habitat, measured by area. In addition to species-specific habitat metrics (Fig. 2), the mean and standard deviation of each metric was calculated across the regional species groups (Supplementary Table 4). To capture model sensitivity to input data, 20 replicate BRTs were fit to 20

different 75% subsamples of the telemetry and pseudo-absence data. Replicate models were predicted across the time-series and habitat metrics were calculated as above. Mean, standard error and coefficient of variation were then calculated for each species, MHW, and habitat metric across the 20 models (see Supplementary methods section 1.6.2, Supplementary Table 5).

The absolute magnitude of change to species' range extent and habitat area varied greatly across species, due in part to differences in occurrence extent (e.g., seabirds have large ranges relative to California sea lions). In order to make MHW impacts comparable across species, range compression/expansion and habitat area loss/gain were expressed as a percent change from mean conditions.

We quantified how species core habitats redistributed across national jurisdictions during MHW events. Version 11 of Exclusive Economic Zone (EEZ) boundaries were used to define national jurisdictions (marineregions.org)[59] (Supplementary Fig. 1). The amount of each species core habitat within the EEZs of Canada, the US, and Mexico, and within the high seas was calculated for each day in August–October of 2000–2020. For each species, the average core habitat area in each jurisdiction was calculated for August–October of each MHW year, and the August–October 2000–2020 baseline was subtracted to obtain the anomaly. Anomalies were expressed as a percent change relative to baseline conditions. Habitat redistribution was summarized as both a percent change for each jurisdiction/MHW, and as the largest percent loss and gain for each species. Mean anomaly, standard error, and coefficient of variation were calculated for each species, MHW, and jurisdiction across the 20 replicate BRT models (see Supplementary methods section 1.7, Supplementary Table 7).

## Data availability

All environmental data are publicly available from CMEMS (https://marine.copernicus.eu/). **Model data**. The telemetry data used in model fitting are available from the Tagging of Pacific Predators project (https://mola.stanford.edu/DataLinks/). Additional telemetry data used in model fitting for black-footed and Laysan albatrosses are available in the BirdLife International Seabird Tracking Database (http://www.seabirdtracking.org). **Validation data**. Species datasets used in model validation are available from eBird (https://ebird.org/home), and the North Pacific Pelagic Seabird Database (https://www.usgs.gov/centers/alaska-science-center/science/north-pacific-pelagic-seabird-database). Additional datasets from the National Observer Program and US surface fishery logbook data were provided under a confidentiality agreement. Requests for observer program data and logbook data can be directed to NOAA (https://www.fisheries.noaa.gov/national/fisheries-observers/national-observer-program and https://www.fisheries.noaa.gov/west-coast/sustainable-fisheries/west-coast-highly-migratory-species-logbooks). NOAA albacore tagging data are not publicly posted at the request of the American Fishermen's Research Foundation (AFRF), who collaborated with NOAA to implement the tagging program. However, these data are freely available for use in research projects through AFRF and NOAA. Further information on how to obtain these data can be directed to B. Muhling (barbara.muhling@noaa.gov). OSU blue whale tagging data are publically available on the Animal Tracking Network (ATN): https://portal.atn.ioos.us/#metadata/507d3d6f-16e7-4f02-91a4-a2142d056c0e/project. Portions of the UCSC elephant seal tagging data are publicly available on from MoveBank (https://www.movebank.org/cms/webapp?gwt_fragment=page=studies,path=study7006760) and ATN (https://portal.atn.ioos.us/#metadata/edc4b2d0-b90d-484b-86da-e1e4212409f6/project). The remaining elephant seal data are not currently public because they are being prepared as part of an in-prep data paper, however, they are available upon reasonable request to D. Costa (costa@ucsc.edu). Source data are provided in this paper.

## Code availability

Code in support of this study is publicly available on GitHub (https://github.com/HeatherWelch/MHW_impacts_top_predators)[60].

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

## Acknowledgements

The primary funding for this study came via a grant from NOAA's Office of Law Enforcement (no grant number; H.W.). Additional support was provided by the National Science Foundation (PRFB 1906332, M.S.S.), NOAA's Climate Program Office (NA22OAR4310560, S.B.), NOAA Climate and Fisheries Adaptation Program (NA20OAR431050, B.A.M.), and by the California Current Integrated Ecosystem Assessment program (no grant number, E.L.H.). Funding for tagging efforts was provided by Naval Facilities Engineering Command Southwest for Commander, U.S. Pacific Fleet (Contract No. N62470-15-D-8006-17F4016 issued to HDR, Inc.), USFWS Migratory Birds (F14PX01125), and NOAA NMFS Alaska Science Center. We thank the Tagging of Pacific Predators (TOPP) scientific teams and all those who supported animal tagging efforts, including captains and crew of vessels that released and recapture tagged animals. We thank our colleagues at Inter-American Tropical Tuna Commission (IATTC) for help with yellowfin tuna, Oregon State University (OSU) for whale tagging, and Grupo Tortuguero and NOAA Southwest Fisheries for turtle and shark research. We thank collaborating TOPP partners and working group leaders for their efforts in coordinating, permitting, and conducting tagging research. We thank

the data management team of TOPP including A. Swithenbank, J. Ganong, and M. Castleton for contributing to TOPP data assembly. We thank J. Childers and Y. Gu for their assistance with the fishing logbook data provision. All animal research was conducted in accordance with appropriate permits (for example, Leatherback turtle Endangered Species Act permit nos. 1159, 1227, and 1596; Marine Mammal Protection Act NMFS permit nos. 14636, 14856, and 19108) and IACUC protocols from Stanford University, Oregon State University, and the University of California. Pacific bluefin tuna work was conducted in Mexican waters with the permission and permits provided by the Mexican Government.

## Author contributions

H.W., S.J.B., E.L.H. conceived the study. H.W. performed the analysis and drafted the manuscript. M.S.S., S.B., S.J.B., M.G.J., B.A.M., T.A.C., M.C., A.W.L., E.L.H. contributed to the interpretation and presentation of results. M.S.S., S.B., B.A.M., S.R.B., S.J.B., B.A.B., M.G.C., D.P.C., R.R.H. F.D.J., C.S.M., D.M.P., S.A.S., L.H.T., J.T.W. provided species datasets for modeling or validation. H.W., L.D. operationalized the models. All authors revised the manuscript.

## Competing interests

The authors declare no competing interests.
