## [Peer Review File · Nature Communications]

Impacts of marine heatwaves on top predator distributions
are variable but predictableREVIEWER COMMENTS

Reviewer #1 (Remarks to the Author):

General comments:

The manuscript "Idiosyncratic effects of marine heatwaves on top predator predicted habitat within the Northeast Pacific" presents novel findings on the redistribution of top predators during extreme environmental events – marine heat waves (MHW). The analysis is based on a large and valuable dataset, producing important results for our understanding of the responses of marine species to abrupt changes in the environment. The main noteworthy result of the manuscript is the variable effect of MHW on core habitat (in terms of direction, distance, percent change range, or area) across species and MHW events, which has further consequences for variable cross-jurisdictional redistributions of these important organisms. The original results presented in this manuscript are based on the unprecedented data set, which considers multiple species and multiple MHW in the standardized analysis. In my opinion, this is the main strength of this work. The analytical approach applied in this analysis is sound, meets the expected standards in the field, and I have just moderate comment to the methodological part. The conclusions and claims are supported by the data and the results. The main results of this manuscript point out significant challenges to the future management of marine fauna in the periods of MHW. Importantly, the Authors extend the work with the presentation of the potential way forward – an operational dynamic ocean management tool that can potentially be used for the prediction of the distribution of the top predators in the near real-time. I have to congratulate the authors for their work which can be a valuable contribution to the existing literature. My comments are mainly composed of suggestions that can improve the presentation of the results or the manuscript's clarity. I would like also to ask for a few clarifications. My only critical points are i) lack of appreciation of uncertainty in the predictions of the core habitat and the calculated metrics of habitat change, ii) the location of centroids of some core habitats on land, which is an artefact of the method, and iii) the operational dynamic ocean management tool is only briefly introduced and lacks serious evaluation in my opinion. Specifically,

i) Prediction errors are present in all habitat models (Barry and Elith, 2006). Booster regression trees (BRT) are powerful and handy tools to predict species' habitats, but they are known to produce different results with different subsets of data. While it could be a demanding task, a deeper evaluation of the source, magnitude, and pattern of errors and uncertainties is essential if the models are to be used transparently in decision-making (Barry and Elith, 2006). It would add significant strength to this manuscript.

One of the potential solutions would be to use replicate BRT models for each species, using independent random samples data, predict the core habitat (and metrics) for each replicate and then calculate uncertainty based on the results of these replicates. The models developed during the cross-validations can be used for this purpose. A similar approach was applied by (Barker et al., 2014) to assess model performance and provide a map of the mean prediction uncertainty across species. Another example can be found in (Woodman et al., 2019). The among-model standard error (SE) of the ensemble predictions and the associated coefficient of variation (CV) were used to calculate map cell-specific uncertainty. But here among-model variance can be used to assess uncertainty in the final metrics.

ii) I understand that some predicted habitat centroids (and ranges) are located on land due to the specific distribution of the species along the coast and in the Gulf of California. But is it valid? I found them unrealistic in some cases. This is my main reservation which might have consequences for the calculated metrics and general results.

iii) I argue that the leave-one-year framework is inadequate for the assessment of the prediction model that supposes to work in a near real-time framework. These are two different time resolutions, and I think that some patterns in the daily distributions can be poorly recreated by the model, even though relatively good performance is observed for year averages.

Specific comments:

-Line 1 (title): While I find "Idiosyncratic" an accurate term, it might be difficult to grasp for some readers. Please consider a change to a more common and straightforward word.

-lines 33-35: Why results for 2015 and the US zone specifically are reported in the abstract? I

would suggest something more general

-Line 179 (Figure 1): I assume that areas under the silhouettes of the studied species are characterized by low densities and low variability, but my preference would be to move the silhouettes out of the maps (or exclude them completely, since the species are listed in the caption) to show undisturbed information. Instead, consider showing EEZ borders and indicating each country/high seas here.

-Lines 108-110: In my understanding, this analysis showed irregular responses across species and MHW, indicating challenges and difficulties of future predictions, rather than serving directly as a basis for informing future management. Please reconsider this sentence.

-Line 127 (Figure 2). It would be easier to match groups if they are indicated on the left side of the figure (next to the species names). In B you might consider adding arrows (actual directions, with N up, S – down, etc.) on top of the colored cells. If uncertainty can be measured for these metrics (see my general comment), it can be presented using transparency (e.g. alpha in ggplot2 package or R), where more transparent cells indicate more uncertain results.

-Line 135: I'm wondering if a second-order analysis can be conducted to assess the variance associated with each component (species/event). It could be a model with two random effects: species (14 levels factor) and MHW event (4 levels factor) fitted to each metric. Modeling directions can be more problematic.

-Line 143: It is not clear if to 22-64% of the initial area or by 22-64%. The same with 335-377% and other percentages in this paragraph.

-Lines 149-153: it is not clear to me if the centers of crosses indicate the center of gravity. I argue that crosses do not indicate process (expansion/compression), but simply a range of core habitat. I suggest a change (here and in extended data) to: "crosses indicate a range of core habitat" and mentioning that the black cross is the climatological mean, while colored crosses are MHW-specific core habitat ranges. And "Inset: percent change in core habitat area relative to climatological conditions."

-Line 167 (Figure 4): I suggest dropping silhouettes in Figure 4 A as they are not especially informative. If the color legend can be moved to the bottom part (below 0), the upper part of this subplot can be used to create an inset with a map of the study area and indicated EEZ/high seas. Then, arrows can be drawn to link the groups of bars and specific areas on the map. If uncertainty in the prediction can be assessed (see my main comment), I would add confidence intervals to the bars (both in Figure 4A and Figure 4B).

-Lines 187-222: Figures and extended data figures are intensively referred to in this part of the manuscript. These results, if essential, should be placed in the Results section and the only main interpretation/conclusion presented in the Discussion section.

-Lines 181-185: Coastal and Southern species are described in this paragraph, but there is no information about Northern species.

-Line 191: Indicate that you refer to the results of this study. "Here," at the beginning?

-Lines 209-212: Is there a reference for this claim?

-Lines 278: Interesting observation based on the presented results is that there are irregular responses of different species across MHW, but they are not unpredictable. It is quite opposite - using a range of environmental variables, including variables other than temperature alone, it is possible to predict species redistributions. I am not sure if this perspective is strong enough in the current version.

-Line 325: why do you refer to extended data figure 2?

-Lines 353-361: day of the year is missing in this paragraph while it was included in extended data table 1. The second sentence of this paragraph repeats what is in the first sentence – delete “including temperature, chlorophyll-a, oxygen, and primary productivity”.

-Lines 363-365: it would be good to provide at least one reference for the methods of pseudo-absence (as in the supplementary materials).

-Line 449: I have the impression that the Authors refer to different figures in different parts of the manuscript, but in some places, it is not essential, while distracting from the main message.

-Line 597: day of the year is not mentioned in the main text.

-Line 598: I suggest keeping the same order of species consequently throughout the manuscript for easier reading. There are two species (white shark, and leatherback turtle) that are missing in this table. If there are no observations in the independent data sets – they should be included in the table with zeros.

-Line 620 (Extended data figure 2): It is difficult to assess the size of points because it is “dominated” by the numerous observations on the elephant seal. I suggest using a log scale to size points accordingly (log-transformed number of observations). Also, since colors are not easy to distinguish (e.g. tunas), I strongly suggest using faceting by species, without grouping. I encourage authors to test the differences in the accuracy between MHW years and non-MHW years. It would provide important information on how well the model predicts species distribution under MHW in comparison to “normal” years. For example, in albacore tuna, the accuracy obtained for the MHW years seems to be above the long-term average.

-Line 633 (Extended Data Figure 3): Some predicted habitat centroids and ranges (e.g. California sea lion) are located on land. I can imagine that it is some kind of artefact of the calculation procedure, e.g. California sea lions are distributed along the coastline and are present in the Gulf of California, thus calculated centroid and range falls on land. The example species selected for the main body of the manuscript are free of these issues. Shouldn't these centroids be truncated to the marine areas before calculating metrics? For example, ArcGIS tools allow the creation of centroid points located inside or contained by the bounds of the input feature (“contained by input features” option), while GDAL/Shapely for Python allows calculating representative point (representative_point() function) which is located inside the polygon. Most importantly, how this affects the calculation of metrics? How this affects the general results?

-Line 643: It would be helpful to indicate in the caption that these anomalies and baseline conditions are calculated for August-October. Please add units for all variables.

-Line 649: bathymetry gained high importance in the models. This variable can be interpreted as a proxy of other conditions (temperature, oxygen, etc.), thus taking over the significance. A short comment might be valuable in the appropriate section.

-Line 652: Overall, it looks OK, but I am not able to properly assess this figure – it has too low resolution.

-Supplementary materials:

-line 56: day of the year – static?

-line 172-174: according to the extended data table 2, there were numerous observations on elephant seal – only yellowfin tuna had inadequate data, but some species were missing in the table, as indicated in the comment above

-line 199: please indicate which measure was used for the assessment of the variable importance

-line 244-248: multivariate/multi-covariate - multivariate methods are not the same as multivariable methods. Multivariate methods have more than one dependent variable, while multivariable methods have one dependent variable and more than one independent variable or covariates. Please correct and use one term consequently.

-line 300: Just a thought: MHW years are included in the climatological mean (2000-2020). If

these years with abnormal environmental conditions (2014, 2015, 2019, 2020) are excluded, the redistribution would be even more pronounced.

-line 310: Since the Southern group is represented by two species, the standard deviation is calculated with two values. Is it meaningful? At least, indicate the number of species included in each group.

-line 349: The Authors provided information here and in the Reporting Summary that R code is available through the GitHub repository, but it was not available during the review (404-page not available). It would help understand some of the steps of the analysis, however, the study doesn't introduce novel algorithms or methods, thus R code review was not crucial.

References:

Barker, N. K. S., Cumming, S. G., and Darveau, M. 2014. Models to predict the distribution and abundance of breeding ducks in Canada. *Avian Conservation and Ecology*, 9.

Barry, S., and Elith, J. 2006. Error and uncertainty in habitat models. *Journal of Applied Ecology*, 43: 413-423.

Woodman, S. M., Forney, K. A., Becker, E. A., DeAngelis, M. L., Hazen, E. L., Palacios, D. M., and Redfern, J. V. 2019. esdm: A tool for creating and exploring ensembles of predictions from species distribution and abundance models. *Methods in Ecology and Evolution*, 10: 1923-1933.

Reviewer #2 (Remarks to the Author):

Given the increasing occurrence and prevalence of marine heatwaves - MHWs - it has been surprising that their impact on the distribution of marine animals has received so little attention. This is a very timely submission, that I believe will be of significance to the field, especially given that it is the first look at how the responses of marine megafauna might vary in response to short-term extreme events.

The results are, hence, quite novel, with the authors predicting distribution shifts during MHW years. The predicted impacts were, unsurprisingly, variable (range expansions/contractions) for the different species considered; but importantly, different MHWs were predicted to have impacted certain species (e.g., shearwaters; sea lions) differently.

It would have been ideal if tracking data spatio-temporally overlapped the MHWs, but the authors built a robust predictive framework to work around the gap in data. The authors did a great job exploring and presenting their results, especially when linking redistribution patterns in relation to political boundaries.

The manuscript is very well written and structured. I have no major comments and am supportive of its publication.

MINOR COMMENTS:

#1 I understand it is quite hard to make figures fit the limited space that journals provide, but Figure 1 should be made bigger since it is especially hard to see the kernel densities. The large number of species outlines for the coastal plot also does not help. This figure should also include a legend for the different species (like Figure 2).

#2 The insets in Figure 3 should indicate what the axis refer to (percent change in area).

#3 L192-194; In the first paragraph of the Discussion the summary of the results is sometimes vague. For example "[...] some predators were predicted to experience near total loss of habitat and range compression, while others were predicted to experience a two-fold habitat increase and significant range expansion." It would be better if authors mention which predators were predicted to experience range expansions/contractions.

#4 L209. "Sooty shearwaters were predicted to expand their range during the 2019 and 2020 events, likely associated with elevated primary productivity in coastal Alaskan waters". It is very minor comment, but Alaska is not shown in Figure 4 under the US EEZ.

I can only congratulate the authors on an excellent manuscript. Also, the supplementary materials are quite thorough, and the authors clearly aware of the caveats of the approach they used.

Reviewer #3 (Remarks to the Author):

Review of “Idiosyncratic effects of marine heatwaves on top predator predicted habitat within the Northeast Pacific” by Welch et al.

This study investigates the effects on top predator species of multiple large-scale marine heatwaves, associated with ‘The Blob’, in the Northeast Pacific from 2014-2019. Telemetry data is used together with selected environmental variables to fit a species distribution model to predict distributions during marine heatwave events. The authors present different responses amongst species but also amongst heatwave events and potential consequences for ecosystem management due to shifting distributions into or out of US waters. The results highlight the urgent need to develop operational ocean management tools.

The manuscript is well written and results presented in a clear way. For full disclosure, this review focuses on the oceanographic aspects and marine heatwave definition of the manuscript. I recommend the manuscript for publication after some (potentially minor) revisions. Detailed comments are outlined below.

Comments

I have a few more general comments about the definitions of marine heatwaves and associated terminology.

The authors do not specifically detect MHWs here but build on existing literature that has identified the 4 big MHW events in the specific years that are referred to here. For completion it might be helpful to add some MHW metrics about those 4 events in the supplementary, e.g. start, end and peak date since marine heatwaves are defined as discrete events. I understand that these might not be as relevant for this study as the focus is on the months with the warmest total temperatures but if the term MHW is used it could be helpful to state them specifically or add a couple of sentences along the lines of what I mentioned above.

Furthermore, the authors speak a few times about averages across each MHW event (e.g. in L299) which can be misleading. Instead, I would say something like Aug-Oct average during each MHW year.

The years 2000-2020 are used as baseline to derive anomalies and the authors refer to it as climatology. I would not call it climatology as that term typically refers to a 30-year average, in particular for MHW studies. Furthermore, the authors justify the use of 2000-2020 with the limitation of the dataset, in particular because of the chlorophyll data, however, SST is available earlier and one could use a more common baseline (e.g. 1982-2010) to derive the anomalies to be more consistent with previous MHW work. Of course, it is under current debate how to best define a MHW (<https://pubmed.ncbi.nlm.nih.gov/37012469/>) and species’ responses will be very different depending on the thermal tolerance and potentially depend stronger on the absolute temperatures instead of anomalies? So, one overall question is, how sensitive are the results to the magnitude of anomalies and it was not clear to me if the habitat model was fit with SST anomalies or absolute SST values?

The authors make the point several times in the text that species impacts will vary across MHW events. It is worth highlighting in the discussion (e.g. L199-230) that MHWs are just a statistical construct, but they

can be caused by very different physical mechanisms, which likely is reflected in other variables too. In the supplementary the authors mention that for example the events in 2019 and 2020 were much shallower than 2014 and 2015, which likely drives different responses. In my opinion it is crucial to discuss the different regional ocean processes associated with different types of marine heatwaves and how these can for example impact oxygen at depth though for example variability in the coastal upwelling. I am aware that a detailed discussion of the physical processes is likely out of scope for this study but still think that it would be important to have a short paragraph in the discussion dedicated to this.

Line-based comments

L135ff: Before talking about the effects of MHWs, I think it would be useful to briefly mention that this study focuses on Aug-Oct only in each MHW year. I am aware that the specifics of each MHW are not the focus of the study but I believe it will help the reader with the interpretation of the results.

Figures

Fig1: This might be personal taste, but I believe maps should always include some meridians and parallels for reference. The authors are likely very familiar with the geography of the region but not all readers will be.

Fig3: Add axis labels or change tick labels to include °N etc.

Figure S1: The panels are very small and have low resolutions which makes it impossible to read the numbers on the colorbar.

Reviewer #1 (Remarks to the Author):

Reviewer 1, Comment 1.

The manuscript "Idiosyncratic effects of marine heatwaves on top predator predicted habitat within the Northeast Pacific" presents novel findings on the redistribution of top predators during extreme environmental events – marine heat waves (MHW). The analysis is based on a large and valuable dataset, producing important results for our understanding of the responses of marine species to abrupt changes in the environment. The main noteworthy result of the manuscript is the variable effect of MHW on core habitat (in terms of direction, distance, percent change range, or area) across species and MHW events, which has further consequences for variable cross-jurisdictional redistributions of these important organisms. The original results presented in this manuscript are based on the unprecedented data set, which considers multiple species and multiple MHW in the standardized analysis. In my opinion, this is the main strength of this work. The analytical approach applied in this analysis is sound, meets the expected standards in the field, and I have just moderate comment to the methodological part. The conclusions and claims are supported by the data and the results. The main results of this manuscript point out significant challenges to the future management of marine fauna in the periods of MHW. Importantly, the Authors extend the work with the presentation of the potential way forward – an operational dynamic ocean management tool that can potentially be used for the prediction of the distribution of the top predators in the near real-time. I have to congratulate the authors for their work which can be a valuable contribution to the existing literature. My comments are mainly composed of suggestions that can improve the presentation of the results or the manuscript's clarity. I would like also to ask for a few clarifications. My only critical points are i) lack of appreciation of uncertainty in the predictions of the core habitat and the calculated metrics of habitat change, ii) the location of centroids of some core habitats on land, which is an artefact of the method, and iii) the operational dynamic ocean management tool is only briefly introduced and lacks serious evaluation in my opinion. Specifically,

Response Reviewer 1, Comment 1. Thank you very much for your thorough read of our manuscript and positive and constructive comments. We particularly appreciate your suggestions to improve Figure 2, and for helping us improve our messaging on the predictability of MHWs.

Reviewer 1, Comment 2.

i) Prediction errors are present in all habitat models (Barry and Elith, 2006). Booster regression trees (BRT) are powerful and handy tools to predict species' habitats, but they are known to produce different results with different subsets of data. While it could be a demanding task, a deeper evaluation of the source, magnitude, and pattern of errors and uncertainties is essential if the models are to be used transparently in decision-making (Barry and Elith, 2006). It would add significant strength to this manuscript. One of the potential solutions would be to use replicate BRT models for each species, using independent random samples data, predict the core habitat (and metrics) for each replicate and then calculate uncertainty based on the results of these replicates. The models developed during the cross-validations can be used for this purpose. A similar approach was applied by (Barker et al., 2014) to assess model performance and provide a map of the mean prediction uncertainty across species. Another example can be found in (Woodman et al., 2019). The among-model standard error (SE) of the ensemble predictions and the associated coefficient of variation (CV) were used to calculate map cell-specific uncertainty. But here among-model variance can be used to assess uncertainty in the final metrics.

Response Reviewer 1, Comment 2. Thank you for bringing up this point and providing guidance for how to evaluate our models' prediction errors. We followed the methods in Barker et al 2014 and created 20 replicate BRT models using independent 75% samples of the telemetry and pseudo-absence data. These 280 models (20 replicates x 14 species) were predicted from 2000-2020, and then habitat metrics were calculated (displacement distance, displacement direction, % change range, % change area). For each metric, species and marine heatwave year, the mean, standard error, and coefficient of variation were

calculated across the 20 model outputs to understand the sensitivity of metrics to the subset of data used in model fitting.

We have created a new supplementary section on habitat metric sensitivity which contains these results in addition to results from our sensitivity analysis on binary vs continuous centroid calculation. Our added text is as follows:

“In addition, we tested the sensitivity of habitat metrics to the data used in model fitting. For each species, telemetry and pseudo-absence data were randomly subsampled into 20 different subsets containing 75% of the data and maintaining the 1:1 ratio of presences to pseudo-absences. A BRT model was fit to each subset (n=280 models; 14 species x 20 data subsets), predicted, and summarized to calculate habitat metrics following methods in Sections 1.5.2., 1.5.6., and 1.6.1. Then, habitat metrics were summarized across the 20 models by mean value, standard error, and coefficient of variation (Table S2). Coefficient of variation was included in addition to standard error because values are standardized by the mean, controlling for larger values having larger standard errors. We found low sensitivity of habitat metrics to data used in model fitting. The average standard errors across species and MHW years were: displacement distance: 9.81 ($\pm 19.2\text{km}$ 95% confidence interval); displacement direction: 2.75 ($\pm 5.39^\circ$ 95% confidence interval); percent change range: 3.52 ($\pm 6.89\%$ 95% confidence interval); and percent change area: 1.68 ($\pm 3.29\%$ 95% confidence interval). Displacement distance had the largest standard error, but its confidence interval indicates that error (19.2km) was less than the size of one 25km pixel. The mean coefficient of variations across species and MHW years were also low: displacement distance: 0.24; displacement direction: 0.18; % change range: 1.14; and % change area: 0.31.”

Further, we added a table to visualize the results and enable comparisons:

Table S2. Sensitivity of habitat metrics to data used in model fitting. For each habitat metric, species, and marine heatwave (MHW), the mean, standard error (SE), and coefficient of variation (CV) were calculated across 20 models fit to random 75% subsets of the telemetry and pseudo-absence data. SE and CV columns have been shaded to ease interpretation such that darkest reds indicate the largest value in each column.

	MHW	Displacement distance			Displacement direction			% change range			% change area		
		Mean	SE	CV	Mean	SE	CV	Mean	SE	CV	Mean	SE	CV
Albacore tuna	2014	390.5	8.8	0.1	-29.4	0.17	0.03	-54.3	2.32	0.19	-40.4	2.27	0.25
	2015	506.5	29.67	0.26	-28	0.4	0.06	-47.9	4.24	0.4	-55.4	2.99	0.24
	2019	188.4	4.92	0.12	130.3	0.69	0.02	-3	5.62	8.47	-37.5	2.39	0.29
	2020	99.2	9.08	0.41	154.4	3.09	0.09	-48.7	2.91	0.27	-43.3	1.69	0.18
Black-footed albatross	2014	155.1	7.51	0.22	-80.7	0.85	0.05	8.9	2.24	1.13	14.6	0.85	0.26
	2015	45.8	7.47	0.73	-49.9	1.2	1.08	-8.6	0.62	0.32	8.9	0.74	0.37
	2019	103.6	2.68	0.12	24	4.24	0.79	-14.7	0.88	0.27	-9.5	0.74	0.35
	2020	130.3	7.14	0.25	73.1	3.84	0.24	-13.2	1.05	0.36	-10.8	0.86	0.36
Blue shark	2014	422.9	22.62	0.24	-26	0.35	0.06	-65.2	0.74	0.05	-34.9	0.95	0.12
	2015	518.7	26.23	0.23	-24.7	0.49	0.09	-83.1	1.92	0.1	-64.8	1.16	0.08
	2019	70.9	10.67	0.67	-177.2	11.3	0.29	-20.6	1.65	0.36	-17.4	1.06	0.27
	2020	159.3	18.22	0.51	-5	10.87	9.76	-14.1	2.38	0.75	-45.9	1.75	0.17
Blue whale	2014	364.4	19.28	0.23	-27.3	0.22	0.04	-72.2	1	0.06	-34.1	1.2	0.15
	2015	467.2	21.63	0.2	-25.6	0.26	0.04	-79.7	1.75	0.1	-35	2.65	0.33
	2019	90.2	6.34	0.31	-150	7.59	0.22	-46.9	2.06	0.19	-12.5	2.21	0.77
	2020	95	12.72	0.58	-127.6	14.18	0.48	-58.7	3.69	0.27	-44.7	2.53	0.25
Bluefin tuna	2014	395.4	2.28	0.03	-33.5	0.08	0.01	-63.3	0.51	0.04	-39.2	0.39	0.04
	2015	636.5	10.3	0.07	-26.8	0.22	0.04	-88.9	1.98	0.1	-94.5	0.3	0.01
	2019	183.3	3.8	0.09	153.2	0.51	0.01	-56.9	2.18	0.17	-47.6	1.38	0.13
	2020	32.9	3.29	0.45	-112.9	10.41	0.41	-42.7	1.73	0.18	-63.3	1.15	0.08
California sea lion	2014	234.6	10.06	0.19	-19.1	0.37	0.09	-91.9	1.38	0.07	-69.2	1.47	0.09
	2015	386.9	5.03	0.06	-14.7	0.27	0.08	-82.9	1.23	0.07	10.7	4.75	1.99
	2019	88.7	9.57	0.48	-15.5	1.16	0.33	184.6	24.27	0.59	152.7	9.55	0.28
	2020	129	7.76	0.27	-20.9	0.76	0.16	338.8	25.21	0.33	191.4	12.76	0.3
Elephant seal	2014	270.6	3.38	0.06	-52.3	0.73	0.06	99.2	1.97	0.09	-29	0.37	0.06
	2015	338.5	3.91	0.05	-59.6	0.66	0.05	10.2	1.24	0.54	-17.6	0.37	0.09
	2019	345.8	4.14	0.05	-76.5	0.36	0.02	29.4	0.78	0.12	69.2	0.96	0.06
	2020	349.1	5.04	0.06	-67.1	0.26	0.02	23.5	0.54	0.1	43.9	1.04	0.11
Laysan albatross	2014	60.7	3.78	0.28	-97	5.19	0.24	-4.5	1.58	1.57	-5.7	1.2	0.94
	2015	44.6	2.17	0.22	-24.2	3.02	0.56	-8	0.46	0.26	-6.9	0.73	0.47
	2019	104.8	4.02	0.17	-126.8	1.66	0.06	-7.9	1.62	0.91	-18.3	0.51	0.12
	2020	92.1	3.01	0.15	-177.1	1.47	0.04	-8.5	0.98	0.51	-11.1	0.72	0.29
Leatherback turtle	2014	100.5	5.98	0.27	-60.6	3.18	0.23	-9.8	1.57	0.71	-10.1	0.39	0.17
	2015	269.3	11.5	0.19	-16.2	3.46	0.96	-31.4	2.75	0.39	-37.3	1.69	0.2
	2019	416.4	23.62	0.25	77.6	1.73	0.1	-32.1	3.57	0.5	-33.5	1.51	0.2
	2020	241.1	14.22	0.26	100.7	4.09	0.18	-27.3	2.06	0.34	-25.1	0.76	0.14
Mako shark	2014	339.8	7.66	0.1	-27.5	0.07	0.01	-1.6	6.77	18.88	-15.9	0.95	0.27
	2015	358.8	11.57	0.14	-24.8	0.33	0.06	12	6.09	2.26	-35.3	1.34	0.17
	2019	69.9	3.79	0.24	140.2	0.76	0.02	9.1	3.52	1.72	6.3	1.45	1.04
	2020	83.5	3.48	0.19	148.2	0.6	0.02	-22.1	1.59	0.32	-2.1	0.69	1.48
Salmon shark	2014	202.9	2.72	0.06	-50.5	0.32	0.03	-31.9	0.61	0.09	7	0.65	0.41
	2015	116.4	4.58	0.18	-42.9	1.11	0.12	-20.9	0.58	0.12	18.7	0.79	0.19
	2019	29	3.38	0.52	-108.1	9.48	0.39	-5.4	0.73	0.6	13.8	0.63	0.2
	2020	83.9	7.19	0.38	119.3	1.54	0.06	0.4	0.83	8.4	12.1	1.06	0.39
Sooty shearwater	2014	233.8	21.76	0.42	9	7.4	3.66	-39.3	5.14	0.58	-30.5	0.96	0.14
	2015	250.1	11.34	0.2	-13.9	2.37	0.76	-53.4	3.67	0.31	-38.3	0.95	0.11
	2019	453.5	27.27	0.27	-71.3	1.87	0.12	280.4	23.02	0.37	-25.8	2.78	0.48
	2020	717.5	39.11	0.24	-55.8	0.75	0.06	295.8	20.94	0.32	-43.6	3.12	0.32
White shark	2014	234.5	7.39	0.14	20.2	1.77	0.39	-19.8	1.27	0.29	-38.4	0.7	0.08
	2015	446.7	8.95	0.09	15.6	0.85	0.24	-40.4	3.52	0.39	-51.5	1.13	0.1
	2019	345.3	17.54	0.23	59.5	1.24	0.09	-44.9	1.13	0.11	-43.1	1.87	0.19
	2020	203.8	10.44	0.23	60.2	1.88	0.14	1	1.7	7.41	-22.5	1.23	0.24
Yellowfin tuna	2014	304.3	3.14	0.05	-30.4	0.08	0.01	-22.3	0.88	0.18	42	1.96	0.21
	2015	346.3	3.62	0.05	-28.8	0.18	0.03	-14.1	1.05	0.33	13.8	1.68	0.55
	2019	8.2	1.16	0.63	-52.2	11.07	0.95	-14.4	0.47	0.14	18	1.13	0.28
	2020	123.7	1.17	0.04	-33.1	0.22	0.03	-28.6	0.67	0.1	13.9	0.83	0.27

Reviewer 1, Comment 3.

ii) I understand that some predicted habitat centroids (and ranges) are located on land due to the specific distribution of the species along the coast and in the Gulf of California. But is it valid? I found them unrealistic in some cases. This is my main reservation which might have consequences for the calculated metrics and general results.

Response Reviewer 1, Comment 3. While a mean centroid is coarse and does not necessarily represent where species actually are, it still shows whether the relative proportions of habitat in different areas changes over time or not. Our method of centroid calculation is common in the literature for marine species¹⁻³. In particular, Pinsky et al. (2018 Science)¹ Fig. 1A shows individual species centroids on land atop the US west coast and the Aleutians due solely to the coastal orography.

To address this point in the text, we have added an explanation in the supplementary materials:

“Due to the convex curvature of the US west coast, centroids for some species were located on land (Extended Data Figure 3). While a centroid is coarse and does not necessarily represent where species actually are, it still captures the relative relocations of preferred habitat among species.”

For clarification, predictions of habitat suitability were truncated to marine areas prior to centroid calculation. In addition, our calculations of species jurisdictional shifts (which require absolute locations of habitat as opposed to relative shifts of habitat) were directly calculated from core habitat as opposed to centroids. There is low variability in centroid location when models are fit on different subsets of data (Reviewer 1, Comment 3), or when centroids are calculated from binary versus continuous habitat suitability (supplementary methods section 1.6.2.):

“We tested the effect of calculating core habitat centroids on binary core habitat (all core habitat pixels assigned a value of 1) versus continuous core habitat (all core habitat pixels maintain values of continuous habitat suitability predictions). This comparison only affects values of habitat displacement (direction and distance) as range extent change and habitat area change do not utilize centroids. We found minimal differences between the two methods of centroid definition, with displacement distance varying on average by 3km, and displacement direction varying on average by 1 degree (Fig. S2).”

Figure S2. Sensitivity of displacement distance (A,B,C) and direction (D,E,F) to core habitat centroids calculated on binary core habitat (all core habitat pixels assigned a value of 1 (A,D)) versus continuous core habitat (all core habitat pixels maintain values of continuous habitat suitability predictions (B,E)). C,F show binary core habitat minus continuous core habitat for habitat displacement distance and direction, respectively.

Reviewer 1, Comment 4.

iii) I argue that the leave-one-year framework is inadequate for the assessment of the prediction model that supposes to work in a near real-time framework. These are two different time resolutions, and I think that some patterns in the daily distributions can be poorly recreated by the model, even though relatively good performance is observed for year averages.

Response Reviewer 1, Comment 4. While we are suggesting these models could be used in a daily dynamic tool, the focus of this paper is instead the general annual patterns in responses to marine heatwaves. For this reason the leave one year out approach is useful to understand how these predictions would remain skillful in years that do not have model-training data, and has become the standard for testing operational models^{4,5}. We have included information about how collecting and assimilating daily observations could be an important next step in the discussion :

“Real-time predictions of species distributions during MHWs could be used to guide observational programs (e.g. field surveys and tagging programs) during anomalous conditions. Importantly, these new observations could be assimilated with distribution model outputs to improve real-time model predictions, a common process in oceanography which is not yet a standard practice in ecology”

In light of this comment, we have added a supplementary analysis of the daily ratio of observed to predicted presences from the telemetry datasets used in model fitting to test how well the model performs on withheld data at this higher temporal resolution. We found that the models did perform well and thus could be operationalized in a daily time step:

“Additionally, we calculated the daily ratio of observed to predicted values at presences from the telemetry datasets used in model fitting (Supplementary methods Figure S3) to evaluate the predictive capabilities of models in a near-real time framework. Each presence was assigned a 1 and then summed

to calculated observed values at presences. To calculate predicted values at presences, model-predicted values at each presence were summed. Ratios of 1 indicate perfect agreement between observed and predicted presences (i.e. all model predictions at presences were 1s), while ratios above and below one indicate under and over prediction, respectively⁶. Ratios showed high alignment between daily observed and predicted presences (mean 1.5 +/- 1.3 SD), indicating strong capacity to generate predictions in a near-real time framework.”

Figure S3. The daily ratio of observed to predicted presences for each of the boosted regression tree models. Ratios of 1 indicate perfect agreement between observed and predicted presences, while ratios above and below one indicate under and over prediction, respectively.

Specific comments:

Reviewer 1, Comment 5.

-Line 1 (title): While I find “Idiosyncratic” an accurate term, it might be difficult to grasp for some readers. Please consider a change to a more common and straightforward word.

Response Reviewer 1, Comment 5. Thank you for the suggestion to make it more accessible. We have changed the title to “Impacts of marine heatwaves on top predator distributions are variable but predictable”.

Reviewer 1, Comment 6.

-lines 33-35: Why results for 2015 and the US zone specifically are reported in the abstract? I would suggest something more general

Response Reviewer 1, Comment 6. Those results were reported because they represent the greatest redistribution, but we agree that this level of detail is not needed in the abstract. We’ve changed this sentence to read:

“MHWs rapidly altered political bio-geographies, with up to 10% of predicted habitat across all species shifting jurisdictions during individual MHWs.”

Reviewer 1, Comment 7.

-Line 179 (Figure 1): I assume that areas under the silhouettes of the studied species are characterized by low densities and low variability, but my preference would be to move the silhouettes out of the maps (or exclude them completely, since the species are listed in the caption) to show undisturbed information. Instead, consider showing EEZ borders and indicating each country/high seas here.

Response Reviewer 1, Comment 7. Thank you for this feedback. We received several comments suggesting edits to this figure, including adding EEZ borders and country names, adding meridians and parallels, and that it was too small to see the kernel densities and contours (Reviewer 1, Comment 7; Reviewer 2, Comment 2; Reviewer 3, Comment 7). However, even making the maps larger, we feel it is still too small to show EEZ borders and still convey the kernels and contours (which is the primary information in this figure):

In lieu of adding EEZs to this figure, we have added an inset map showing EEZs to Figure 4, as suggested by Reviewer 1, Comment 13. As requested, we have made the maps in Figure 1 bigger, and have removed the species outlines:

Reviewer 1, Comment 8.

-Lines 108-110: In my understanding, this analysis showed irregular responses across species and MHW, indicating challenges and difficulties of future predictions, rather than serving directly as a basis for informing future management. Please reconsider this sentence.

Response Reviewer 1, Comment 8. Thank you for flagging this lack of clarity in our messaging. We have rewritten the discussion section on early warning systems to better communicate how our analysis serves as a basis for informing future management (in response to Reviewer 1, Comment 18.):

“Our results show a wide range of species impacts across MHW events and jurisdictions. This diversity of responses poses a daunting management challenge: how to plan ahead and respond swiftly to MHW-driven species redistribution. Our results indicate that species responses to MHWs are highly variable yet predictable: our models performed well through extensive validation across space, time, and on novel data (Extended Data Figure 2 and Table 3). The high variability of species responses to MHWs suggests that we cannot assume future MHWs will impact species in the same way as past events. However, high predictability indicates that species responses to future MHWs could be predicted in real-time to provide accurate information on impacts. This combination of high variability and predictability is also seen in hurricanes – future hurricane paths cannot be extrapolated from past paths, yet they can be predicted in real-time to provide accurate information on risk.”

Reviewer 1, Comment 9.

-Line 127 (Figure 2). It would be easier to match groups if they are indicated on the left side of the figure (next to the species names). In B you might consider adding arrows (actual directions, with N up, S – down, etc.) on top of the colored cells. If uncertainty can be measured for these metrics (see my general comment), it can be presented using transparency (e.g. alpha in ggplot2 package of R), where more transparent cells indicate more uncertain results.

Response Reviewer 1, Comment 9. Thank you, it's a good idea to add direction arrows and move species groups to the left side of the figure, we have made both of these changes:

Figure 2. Predicted impacts on top predator habitat within (columns, e.g. 2014) and among (rows, e.g. White shark) marine heatwave events measured using four metrics: A. displacement distance (kilometers), B. displacement direction (degrees, where 0/360 is north (N), 90 is east (E), 180 is south (S), and 270 is west (W)), C. range compression or expansion (percent change relative to baseline conditions), D. habitat area gain or loss (percent change relative to baseline conditions). All metrics were calculated from August-October in each MHW year relative to baseline conditions (August-October 2000-2020), see Table S2 for an analysis of metric uncertainty. Northern, Coastal, and Southern regional groupings indicate the geographies where the majority of the species telemetry data occurs.

However, we have elected to not use alpha to indicate uncertainty simply because it was extremely difficult to discern the original intent of the figures. Visualizing uncertainty using alpha masks the relationship between the tile and legend colors, making it challenging to communicate magnitudes. For example, in the plot below, the left panel doesn't use alpha, while the right panel maps alpha to uncertainty. In addition, because the habitat metrics vary across different magnitudes (e.g. distance ranges (0-700 km) versus percent change range (-100% to 370%), the standard errors aren't comparable across habitat metrics (larger numbers will have larger standard errors and lower alphas, penalizing habitat metrics that vary over larger magnitudes). Lastly, to include uncertainty in this figure, we would need to change the habitat metric values currently displayed (outputs from the full model containing 100% of the data) to the mean habitat metrics averaged across the 20 model replicates of 75% data (the mean described by standard error). However, we feel that outputs from the full model containing 100% of the data are the most accurate, given that they were derived from the most data.

Instead, we have added Table S2 that shows, for each species, MHW and habitat metric, the mean, standard error, and coefficient of variation across 20 replicate models, and point to it in the caption of Figure 2 (see Response Reviewer 1, Comment 2.).

Reviewer 1, Comment 10.

-Line 135: I'm wondering if a second-order analysis can be conducted to assess the variance associated with each component (species/event). It could be a model with two random effects: species (14 levels factor) and MHW event (4 levels factor) fitted to each metric. Modeling directions can be more problematic.

Response Reviewer 1, Comment 10. We have added a new section and Table S3 to the supplementary information following this suggestion (section 1.6.3.). As you mention, it is challenging to build models for direction, which is a circular variable and must be handled differently than the other metrics. There are several functions we explored for this purpose (`lm.circular()` and `bpmr()`), but we had concerns about how to appropriately structure and interpret circular linear models, and how to compare these models with models for the other habitat metrics. As a result, we have elected to not model direction, but have added the following text and table for the other habitat metrics following your suggestions:

“We explored the extent to which variability in continuous habitat metrics was associated with species vs MHWs using linear mixed effect models. For each habitat metric (distance, percent change range, percent change area), we built two models with one fixed effect and one random effect, e.g. for percent change area:

Mod1: $lmer(\text{percent change area} \sim \text{MHW} + (1|\text{species}), \text{data} = \text{dat})$
Mod2: $lmer(\text{percent change area} \sim \text{species} + (1|\text{MHW}), \text{data} = \text{dat})$

In Mod1, MHW is fixed and species is random, and in Mod2, MHW is random and species is fixed. Next, we compared the marginal R^2 values for each model, i.e. the R^2 considering only the fixed effect and ignoring the random effect. This comparison allowed us to determine if MHW explains more variance while controlling for species (Mod1), or if species explains more variance while controlling for MHW (Mod2) (Table S3).

Table S3. Linear mixed models for percent change range, percent change area, and distance habitat metrics. Each model contained one random effect (either species or marine heatwave (MHW)) and one fixed effect. Marginal R² values indicate the variance explained by the fixed effect while controlling for the random effect.

Metric	Random effect	Fixed effect	Marginal R²	Tests
Percent change range	Species	MHW	0.1	The variance explained by MHW
Percent change range	MHW	Species	0.31	The variance explained by species
Percent change area	Species	MHW	0.04	The variance explained by MHW
Percent change area	MHW	Species	0.55	The variance explained by species
Distance	Species	MHW	0.13	The variance explained by MHW
Distance	MHW	Species	0.25	The variance explained by species

Results indicate that species explains more of the variance than MHW for percent change range, percent change area, and distance habitat metrics.”

Reviewer 1, Comment 11.

-Line 143: It is not clear if to 22-64% of the initial area or by 22-64%. The same with 335-377% and other percentages in this paragraph.

Response Reviewer 1, Comment 11. The percentages in this paragraph are percent increases and decreases, i.e. the difference between the final and initial values, expressed as a percentage of the initial value. We have clarified this in the caption of Fig. 2 which is referenced in this paragraph, and now report percentages as “% increase” and “% decrease”.

Reviewer 1, Comment 12.

-Lines 149-153: it is not clear to me if the centers of crosses indicate the center of gravity. I argue that crosses do not indicate process (expansion/compression), but simply a range of core habitat. I suggest a change (here and in extended data) to: “crosses indicate a range of core habitat” and mentioning that the black cross is the climatological mean, while colored crosses are MHW-specific core habitat ranges. And “Inset: percent change in core habitat area relative to climatological conditions.”

Response Reviewer 1, Comment 12. We agree that our wording was not clear, and have edited the caption of Figure 3 and Extended Data Figure 3 to better distinguish pattern from process:

“Large maps: arrows indicate predicted habitat displacement from the center of gravity in each MHW (center of colored crosses) relative to the center of gravity during baseline conditions (center of black cross). Crosses indicate predicted longitudinal and latitudinal range extents during each MHW and during baseline conditions (colored and black crosses, respectively); an increase in cross size during a MHW compared to baseline conditions indicates range expansion with the converse representing compression. Inset: percent change habitat area relative to baseline conditions.”

Reviewer 1, Comment 13.

-Line 167 (Figure 4): I suggest dropping silhouettes in Figure 4 A as they are not especially informative. If the color legend can be moved to the bottom part (below 0), the upper part of this subplot can be used to create an inset with a map of the study area and indicated EEZ/high seas. Then, arrows can be drawn to link the groups of bars and specific areas on the map. If uncertainty in the prediction can be assessed (see my main comment), I would add confidence intervals to the bars (both in Figure 4A and Figure 4B).

Response Reviewer 1, Comment 13. We appreciate your suggestion to drop the silhouettes from Figure 4A and include a map of the study area indicating jurisdictional boundaries, and have made both changes. We elected to not draw arrows from each bar group to each map jurisdiction because it felt too busy.

Figure 4. Cross-jurisdictional shifts in predators' predicted habitats during marine heatwaves. A. Total loss and gain of species habitat area across the US, Mexican (MEX), and Canadian (CAN) Exclusive Economic Zones (EEZs), and the high seas during each MHW. B. Largest loss and gain of habitat for each predator in any MHW event (x-axis text color indicates species regional groupings: Southern (green), Coastal (black), Northern (blue)). Percent change in habitat area is calculated relative to baseline conditions (2000-2020), see Table S4 for an analysis of uncertainty in cross-jurisdictional shifts.

In lieu of adding uncertainty bars to this figure, we have created a new supplementary Table S4 and now point to it in the Figure 4 caption. To include uncertainty in this figure, we would need to change the anomaly values currently displayed (outputs from the full model containing 100% of the data) to the mean anomaly values across the 20 model replicates of 75% data (the mean described by standard error). However, we feel that outputs from the full model containing 100% of the data are the most accurate, given that they were derived from the most data.

Here is our new supplementary text and table describing the analysis of jurisdictional uncertainty:

“Similar to the habitat metrics, we tested the sensitivity of species core habitat redistribution across jurisdictions to the data used in model fitting by fitting and predicting 20 replicate BRT models (Section

1.6.2.). Habitat redistributions were calculated as above, and summarized across the 20 models by mean anomaly, standard error, and coefficient of variation (Table S4). The average standard errors and coefficients of variations across species, MHW years, and jurisdictions were low: 0.0031 ($\pm 0.0061\%$ 95% confidence interval) and 1.47 for standard error and coefficient of variation, respectively.

Table S4. Sensitivity of species core habitat redistribution across jurisdictions to data used in model fitting. For each jurisdiction, species, and marine heatwave (MHW), the mean anomaly, standard error (SE), and coefficient of variation (CV) were calculated across 20 models fit to random 75% subsets of the telemetry and pseudo-absence data. SE and CV columns have been shaded to ease interpretation such that darkest reds indicate the largest value in each column.”

	Canadian EEZ				High seas			US EEZ			Mexican EEZ		
	MHW	Mean	SE	CV	Mean	SE	CV	Mean	SE	CV	Mean	SE	CV
Albacore tuna	2014	0.00%	NA	NA	0.31%	0.0008	1.1	17.06%	0.0046	0.12	-17.37%	0.0048	0.12
	2015	0.00%	NA	NA	-2.57%	0.0019	0.32	20.24%	0.0062	0.14	-17.67%	0.0050	0.13
	2019	0.00%	NA	NA	-2.28%	0.0021	0.41	-6.61%	0.0077	0.52	8.89%	0.0082	0.41
	2020	0.00%	NA	NA	-3.01%	0.0027	0.4	7.60%	0.0139	0.82	-4.59%	0.0125	1.22
Black-footed albatross	2014	-1.56%	0.0010	0.29	1.58%	0.0027	0.75	-0.02%	0.0030	74.39	0.00%	NA	NA
	2015	-0.39%	0.0011	1.26	4.38%	0.0031	0.31	-3.99%	0.0031	0.35	0.00%	NA	NA
	2019	1.23%	0.0013	0.47	0.25%	0.0061	11.12	-1.48%	0.0065	1.98	0.00%	NA	NA
	2020	1.50%	0.0021	0.63	2.07%	0.0052	1.14	-3.57%	0.0061	0.77	0.00%	NA	NA
Blue shark	2014	0.75%	0.0025	1.48	-0.48%	0.0017	1.55	9.89%	0.0027	0.12	-10.15%	0.0024	0.11
	2015	-0.11%	0.0038	15.42	-0.75%	0.0021	1.25	11.45%	0.0041	0.16	-10.59%	0.0027	0.11
	2019	-2.30%	0.0028	0.55	-0.77%	0.0011	0.64	6.39%	0.0045	0.31	-3.33%	0.0071	0.95
	2020	-1.60%	0.0032	0.9	-0.81%	0.0013	0.73	6.77%	0.0039	0.26	-4.35%	0.0065	0.67
Blue whale	2014	0.74%	0.0013	0.76	-2.81%	0.0100	1.59	17.18%	0.0216	0.56	-15.12%	0.0122	0.36
	2015	1.18%	0.0010	0.37	-1.99%	0.0011	0.25	15.32%	0.0082	0.24	-14.50%	0.0077	0.24
	2019	-1.07%	0.0016	0.67	-2.42%	0.0102	1.89	15.87%	0.0215	0.6	-12.38%	0.0122	0.44
	2020	-0.14%	0.0045	14.03	-2.06%	0.0103	2.24	12.86%	0.0230	0.8	-10.66%	0.0127	0.53
Bluefin tuna	2014	0.00%	NA	NA	-0.19%	0.0002	0.43	10.90%	0.0044	0.18	-10.71%	0.0043	0.18
	2015	0.00%	NA	NA	-0.19%	0.0002	0.43	10.78%	0.0036	0.15	-10.59%	0.0034	0.15
	2019	0.00%	NA	NA	-0.16%	0.0001	0.24	0.89%	0.0060	3.03	-0.73%	0.0060	3.69
	2020	0.00%	NA	NA	-0.19%	0.0002	0.39	4.70%	0.0041	0.39	-4.51%	0.0041	0.41
California sea lion	2014	0.00%	NA	NA	-0.82%	0.0012	0.63	0.92%	0.0014	0.67	-0.09%	0.0006	2.77
	2015	0.00%	NA	NA	-0.81%	0.0011	0.62	0.90%	0.0013	0.65	-0.09%	0.0006	2.77
	2019	0.00%	NA	NA	-0.15%	0.0009	2.82	0.24%	0.0011	2.11	-0.09%	0.0006	2.77
	2020	0.00%	NA	NA	-0.67%	0.0010	0.69	0.75%	0.0012	0.73	-0.08%	0.0006	3
Elephant seal	2014	2.34%	0.0006	0.12	-9.23%	0.0015	0.07	7.22%	0.0015	0.09	-0.33%	0.0004	0.49
	2015	2.10%	0.0014	0.29	-2.49%	0.0025	0.45	1.00%	0.0014	0.63	-0.62%	0.0003	0.24
	2019	0.70%	0.0007	0.43	2.96%	0.0013	0.19	-2.60%	0.0007	0.11	-1.07%	0.0002	0.07
	2020	4.38%	0.0010	0.1	-0.44%	0.0013	1.34	-2.99%	0.0007	0.11	-0.94%	0.0002	0.08
Laysan albatross	2014	-0.01%	0.0000	0.6	-0.61%	0.0101	7.41	0.62%	0.0101	7.31	0.00%	NA	NA
	2015	-0.01%	0.0000	0.64	-0.30%	0.0052	7.63	0.31%	0.0052	7.36	0.00%	NA	NA
	2019	-0.01%	0.0000	0.59	5.63%	0.0086	0.69	-5.62%	0.0087	0.69	0.00%	NA	NA
	2020	-0.01%	0.0000	0.65	10.28%	0.0075	0.33	-10.27%	0.0075	0.33	0.00%	NA	NA
Leatherback turtle	2014	0.00%	NA	NA	3.73%	0.0052	0.62	-1.02%	0.0031	1.36	-2.71%	0.0022	0.36
	2015	0.00%	NA	NA	-2.43%	0.0104	1.92	5.82%	0.0122	0.94	-3.39%	0.0020	0.27
	2019	0.00%	NA	NA	-12.01%	0.0043	0.16	7.96%	0.0044	0.25	4.05%	0.0017	0.18
	2020	0.00%	NA	NA	-8.00%	0.0083	0.46	6.58%	0.0073	0.5	1.42%	0.0014	0.43
Mako shark	2014	0.00%	NA	NA	0.37%	0.0002	0.29	13.99%	0.0043	0.14	-14.35%	0.0045	0.14
	2015	0.00%	NA	NA	0.79%	0.0006	0.34	14.48%	0.0062	0.19	-15.27%	0.0065	0.19
	2019	0.00%	NA	NA	-0.04%	0.0002	2.91	-2.66%	0.0015	0.25	2.70%	0.0016	0.26
	2020	0.00%	NA	NA	0.13%	0.0003	1.06	-2.19%	0.0019	0.4	2.06%	0.0018	0.4
Salmon shark	2014	-0.56%	0.0006	0.51	-0.18%	0.0003	0.76	0.75%	0.0008	0.45	-0.01%	0.0000	0.72
	2015	4.42%	0.0018	0.18	-0.21%	0.0003	0.72	-4.20%	0.0018	0.19	-0.01%	0.0000	0.72
	2019	1.21%	0.0015	0.54	0.04%	0.0002	2.09	-1.24%	0.0015	0.53	-0.01%	0.0000	0.72
	2020	3.16%	0.0022	0.31	0.32%	0.0003	0.39	-3.46%	0.0022	0.28	-0.01%	0.0000	0.72
Sooty shearwater	2014	6.00%	0.0023	0.17	-0.29%	0.0016	2.45	-0.94%	0.0033	1.56	-4.77%	0.0023	0.21
	2015	3.76%	0.0029	0.35	-0.31%	0.0014	2.05	1.69%	0.0034	0.9	-5.15%	0.0019	0.16
	2019	-2.73%	0.0034	0.55	0.14%	0.0024	7.85	6.62%	0.0046	0.31	-4.02%	0.0017	0.19
	2020	1.22%	0.0061	2.25	0.71%	0.0031	1.92	1.17%	0.0095	3.63	-3.10%	0.0017	0.25
White shark	2014	0.00%	NA	NA	-15.50%	0.0039	0.11	15.30%	0.0038	0.11	0.19%	0.0004	0.83
	2015	0.00%	NA	NA	-29.67%	0.0092	0.14	29.69%	0.0092	0.14	-0.02%	0.0000	0.78
	2019	0.00%	NA	NA	-33.15%	0.0118	0.16	33.17%	0.0118	0.16	-0.02%	0.0000	1.07
	2020	0.00%	NA	NA	-17.71%	0.0053	0.13	17.55%	0.0053	0.13	0.16%	0.0001	0.41
Yellowfin tuna	2014	0.00%	NA	NA	0.29%	0.0003	0.45	22.44%	0.0035	0.07	-22.73%	0.0035	0.07
	2015	0.00%	NA	NA	0.21%	0.0002	0.53	30.31%	0.0033	0.05	-30.52%	0.0033	0.05
	2019	0.00%	NA	NA	0.03%	0.0002	2.95	-10.81%	0.0020	0.08	10.78%	0.0021	0.09
	2020	0.00%	NA	NA	-0.12%	0.0001	0.2	-2.81%	0.0022	0.36	2.93%	0.0022	0.34

Reviewer 1, Comment 14.

-Lines 187-222: Figures and extended data figures are intensively referred to in this part of the manuscript. These results, if essential, should be placed in the Results section and the only main interpretation/conclusion presented in the Discussion section.

Response Reviewer 1, Comment 14. We have removed redundant extended data figure references, and limited our references to main text figures in the Discussion.

Reviewer 1, Comment 15.

-Lines 181-185: Coastal and Southern species are described in this paragraph, but there is no information about Northern species.

Response Reviewer 1, Comment 15. We now include the following text on northern species: “Northern species experienced the largest redistributions during the 2020 event, with 14% of salmon shark habitat predicted to shift into the Canadian EEZ.”

Reviewer 1, Comment 16.

-Line 191: Indicate that you refer to the results of this study. “Here,” at the beginning?

Response Reviewer 1, Comment 16. Good point, we have now added this for clarification..

Reviewer 1, Comment 17.

-Lines 209-212: Is there a reference for this claim?

Response Reviewer 1, Comment 17. We have added a new supplementary figure showing the conditions of variables beyond temperature to support this claim:

A. Sea surface temperature

B. Oxygen at 200m

C. Mean primary productivity in the upper 200m

D. Log chlorophyll-a

Figure S1. Anomalies of sea surface temperature ($^{\circ}\text{C}$; A), oxygen concentration at 200m (mmol m^{-3} ; B), mean primary productivity in the upper 200m (mg m^{-3} ; C), and log chlorophyll-a (mg m^{-3} ; D) during each marine heatwave event. Anomalies were calculated from August-October in each year relative to a 2000-2020 August-October baseline.

We have also included a couple sentences on regional oceanographic processes in the section to help explain sooty shearwater and elephant seal responses:

“However, temperature anomalies were fairly consistent across the 75th kernels of sooty shearwaters and elephant seals during the four MHW events, yet each species experienced markedly different predicted impacts among MHWs. Sooty shearwaters were predicted to expand their range during the 2019 and 2020 events, likely associated with elevated primary productivity in coastal Alaskan waters (Fig. S1). Elephant seals were predicted to experience large habitat loss during the 2014 and 2015 events, likely due to a reduction of primary productivity and subsurface oxygen in the north-central Pacific (Fig. S1). The deep warm water anomalies of the 2014 and 2015 events decreased the solubility of oxygen and led

to a near-surface decline in oxygen concentration⁷. The 2019 and 2020 events, in contrast, had much shallower temperature anomalies, and likely had a weaker impact on local biogeochemical signals and productivity. In general, variations in winds, air-sea gas exchange, circulation patterns and water column stratification can lead to very different physical and biogeochemical conditions among MHW events. The idiosyncratic responses of species to MHWs reflect these variable conditions and differing species-environment relationships (Extended Data Figure 5)."

Reviewer 1, Comment 18.

-Lines 278: Interesting observation based on the presented results is that there are irregular responses of different species across MHW, but they are not unpredictable. It is quite opposite - using a range of environmental variables, including variables other than temperature alone, it is possible to predict species redistributions. I am not sure if this perspective is strong enough in the current version.

Response Reviewer 1, Comment 18. We appreciate this comment, we've had some trouble articulating that while it is not possible to extrapolate future MHW impacts based on historical impacts, it is possible to predict impacts in near real-time. We've rewritten this section using a more familiar example - hurricanes, which are highly variable but predictable:

"Our results show a wide range of species impacts across MHW events and jurisdictions. This diversity of responses poses a daunting management challenge: how to plan ahead and respond swiftly to MHW-driven species redistribution. Our results indicate that species responses to MHWs are highly variable yet predictable: our models performed well through extensive validation across space, time, and on novel data (Extended Data Figure 2 and Table 3). The high variability of species responses to MHWs suggests that we cannot assume future MHWs will impact species in the same way as past events. However, high predictability indicates that species responses to future MHWs could be predicted in real-time to provide accurate information on impacts. This combination of high variability and predictability is also seen in hurricanes – future hurricane paths cannot be extrapolated from past paths, yet they can be predicted in real-time to provide accurate information on risk."

Reviewer 1, Comment 19.

-Line 325: why do you refer to extended data figure 2?

Response Reviewer 1, Comment 19. Thank you, this was an error caused by a reordering of the figures. We have removed this figure reference.

Reviewer 1, Comment 20.

-Lines 353-361: day of the year is missing in this paragraph while it was included in extended data table 1. The second sentence of this paragraph repeats what is in the first sentence – delete "including temperature, chlorophyll-a, oxygen, and primary productivity".

Response Reviewer 1, Comment 20. Thank you, we have now included day of year, and removed the repetitive text.

Reviewer 1, Comment 21.

-Lines 363-365: it would be good to provide at least one reference for the methods of pseudo-absence (as in the supplementary materials).

Response Reviewer 1, Comment 21. This is a good point. We have included the following reference: Barbet-Massin, Morgane, et al. "Selecting pseudo-absences for species distribution models: How, where and how many?." *Methods in ecology and evolution* 3.2 (2012): 327-338.

Reviewer 1, Comment 22.

-Line 449: I have the impression that the Authors refer to different figures in different parts of the manuscript, but in some places, it is not essential, while distracting from the main message.

Response Reviewer 1, Comment 22. We have removed these figure references.

Reviewer 1, Comment 23.

-Line 597: day of the year is not mentioned in the main text.

Response Reviewer 1, Comment 23. We now include day of year in the Online Methods.

Reviewer 1, Comment 24.

-Line 598: I suggest keeping the same order of species consequently throughout the manuscript for easier reading. There are two species (white shark, and leatherback turtle) that are missing in this table. If there are no observations in the independent data sets – they should be included in the table with zeros.

Response Reviewer 1, Comment 24. We have added white shark and leatherback to the table and indicated that they have no records. We have also reordered Extended Data items Table 2, Table 3, Figure 2, Figure 3, Figure 5, and Figure 6 to list species alphabetically. Main text Figures 2, 4 and Extended Data Figure 4 have different orders to better communicate the regional species groups and the magnitudes of cross-jurisdictional shifts that are presented in the figures.

Reviewer 1, Comment 25.

-Line 620 (Extended data figure 2): It is difficult to assess the size of points because it is “dominated” by the numerous observations on the elephant seal. I suggest using a log scale to size points accordingly (log-transformed number of observations).

Also, since colors are not easy to distinguish (e.g. tunas), I strongly suggest using faceting by species, without grouping.

Response Reviewer 1, Comment 25. We have made both of these changes:

Reviewer 1, Comment 26.

-Line 620 (Extended data figure 2): I encourage authors to test the differences in the accuracy between MHW years and non-MHW years. It would provide important information on how well the model predicts species distribution under MHW in comparison to “normal” years. For example, in albacore tuna, the accuracy obtained for the MHW years seems to be above the long-term average.

Response Reviewer 1, Comment 26. We have performed a t-test comparing AUC during MHW vs normal years and added the results to the figure caption:

“Across all species, there was not a significant difference in AUC between marine heatwave years (mean= 0.78) and normal years (mean=0.78); $t(52.20) = -0.27$, $p\text{-value} = 0.791$.”

Reviewer 1, Comment 27.

-Line 633 (Extended Data Figure 3): Some predicted habitat centroids and ranges (e.g. California sea lion) are located on land. I can imagine that it is some kind of artefact of the calculation procedure, e.g. California sea lions are distributed along the coastline and are present in the Gulf of California, thus calculated centroid and range falls on land. The example species selected for the main body of the manuscript are free of these issues. Shouldn't these centroids be truncated to the marine areas before calculating metrics? For example, ArcGIS tools allow the creation of centroid points located inside or contained by the bounds of the input feature (“contained by input features” option), while GDAL/Shapely for Python allows calculating representative point (representative_point() function) which is located inside the polygon. Most importantly, how this affects the calculation of metrics? How this affects the general results?

Response Reviewer 1, Comment 27. Please see our full response in Reviewer 1, Comment 3 - our method is commonly used in the scientific literature. In addition, we investigated the two tools suggested here as well as a similar R tool and found that none were suitable for our analysis. Our analysis includes (for a given species and day): a raster of core habitat patches (A), and optionally the minimum bounding polygon which we used to constrain the species predictions (B).

Tool 1. “Contained by input features option” in ArcGIS.

We could convert our raster (A) to points to work with this tool, which requires point, line or polygon data. If we checked the “inside” parameter to ensure the output point fell within the spatial domain of the input (note this wouldn't be (B), it would be the points in (A)), the input point that is nearest to the centroid would be selected as the output: “The output point will be coincident with one of the points in the multipoint”. In other words, the pixel containing species core habitat that is closest to the centroid would be selected, which is not the metric we are aiming to calculate. <https://pro.arcgis.com/en/pro-app/latest/tool-reference/data-management/feature-to-point.htm>.

Tool 2. “representative_point()” in GDAL/Shapely for Python

This tool will find the representative point located within a polygon by shifting the true centroid to a location inside a polygon that balances polygon complexity and shape. This means that for polygons that are irregularly shaped such as the minimum bounding polygons (B), portions of the polygon that are larger will pull the representative point towards them. This functionality would change our results from describing the center of gravity of species core habitat (A) to the center of gravity of the polygon used to constrain predictions (B). Furthermore, this tool finds the representative center of a polygon, as opposed to the representative center of a point cluster that is contained within a polygon.

Tool 3. st_point_on_surface() in R

This tool works similarly to Tool 2 - for a line or polygon, it will find the center of gravity that falls within or on the input surface. However, we are aiming to find the centroid of (A) that falls within (B). If we use points as an input (by converting the raster (A) to point), it returns all input points (which are the

center of gravity of themselves) as opposed to one point summarizing all points. Like tool 2, this tool finds the representative center of a polygon or line, as opposed to the representative center of a point cluster that is contained within a polygon.

In summary, none of these tools achieve the desired functionality, which is to summarize the center of gravity of (A) while constraining it within polygon (B). Even if the desired functionality was possible, we feel it could bias results for species for which centroids appear on land and need to be shifted vs species for which centroids appear on water and don't need to be shifted. We feel the simplest and most defensible (given its prevalence in the literature) way forward is to find geometric centroids for all species, with the understanding that values do not describe the absolute locations of habitat but rather how proportions of habitat in different locations change over time.

Reviewer 1, Comment 28.

-Line 643: It would be helpful to indicate in the caption that these anomalies and baseline conditions are calculated for August-October. Please add units for all variables.

Response Reviewer 1, Comment 28. Thank you, we have updated the caption to read:

“Anomalies of Sea surface temperature ($^{\circ}\text{C}$; A), log chlorophyll-a (mg m^{-3} ; B), oxygen at 200m (mmol m^{-3} ; C), and mean primary productivity in the upper 200m (mg m^{-3} ; D) measured over the 75% kernel of each species. Anomalies were calculated from August-October in each year relative to a 2000-2020 August-October baseline.”

Reviewer 1, Comment 29.

-Line 649: bathymetry gained high importance in the models. This variable can be interpreted as a proxy of other conditions (temperature, oxygen, etc.), thus taking over the significance. A short comment might be valuable in the appropriate section.

Response Reviewer 1, Comment 29. It's true that many of our variables are correlated (correlation plot below). We've included a comment in the supplements that BRTs are robust to collinearity effects⁸, but that collinearity can obscure relative importance:

“BRTs are a common machine learning model, popularized by their ability to fit complex nonlinear relationships, automatic handling of collinearity effects, and their robustness to wide varieties of data types and distributions. Relative importance captures each variable's influence on the response⁸, and reflects the number of times each variable is selected for splitting across all trees. Importance is scaled across all variables such that the sum adds to 100, producing relative importance, where higher values indicate stronger contributions. However, relative importance may not accurately reflect variable influence when variables are highly correlated.”

Reviewer 1, Comment 30.

-Line 652: Overall, it looks OK, but I am not able to properly assess this figure – it has too low resolution.
Response Reviewer 1, Comment 30. Apologies, we have remade it at a higher resolution.

-Supplementary materials:

Reviewer 1, Comment 31.

-line 56: day of the year – static?
Response Reviewer 1, Comment 31. We have removed the word “static”.

Reviewer 1, Comment 32.

-line 172-174: according to the extended data table 2, there were numerous observations on elephant seal – only yellowfin tuna had inadequate data, but some species were missing in the table, as indicated in the comment above
Response Reviewer 1, Comment 32. Thank you for catching this, we received the elephant sea data last and failed to update this sentence to reflect the new data. We did not have adequate validation data for yellowfin tuna, white shark, or leatherback turtle and have updated this sentence and extended data table 2 accordingly.

Reviewer 1, Comment 33.

-line 199: please indicate which measure was used for the assessment of the variable importance
Response Reviewer 1, Comment 33. We have added the following text:

“Relative importance captures each variable’s influence on the response⁸, and reflects the number of times each variable is selected for splitting across all trees. Importance is scaled across all variables

such that the sum adds to 100, producing relative importance, where higher values indicate stronger contributions.”

Reviewer 1, Comment 34.

-line 244-248: multivariate/multi-covariate - multivariate methods are not the same as multivariable methods. Multivariate methods have more than one dependent variable, while multivariable methods have one dependent variable and more than one independent variable or covariates. Please correct and use one term consequently.

Response Reviewer 1, Comment 34. Thank you for noticing this inconsistency. We’ve replaced “multivariate/multi-covariate” with “multivariable”.

Reviewer 1, Comment 35.

-line 300: Just a thought: MHW years are included in the climatological mean (2000-2020). If these years with abnormal environmental conditions (2014, 2015, 2019, 2020) are excluded, the redistribution would be even more pronounced.

Response Reviewer 1, Comment 35. While it is true that excluding the warmest years from the climatology would separate them even more from mean conditions, we include all years in the climatology for consistency with the MHW literature (and with calculation of anomalies more generally)⁹⁻¹¹.

Reviewer 1, Comment 36.

-line 310: Since the Southern group is represented by two species, the standard deviation is calculated with two values. Is it meaningful? At least, indicate the number of species included in each group.

Response Reviewer 1, Comment 36 We have indicated the number of species in each group:

Species Group	MHW	Displacement		Percent change	
		Distance	Direction	Range	Area
Northern (3 species)	2014	182km ± 95	West-NW (291° ± 27)	12% ± 60	8% ± 33
	2015	137km ± 143	NW-North (328° ± 23)	-12% ± 18	15% ± 40
	2019	161km ± 118	West-NW (306° ± 60)	-3% ± 25	21% ± 43
	2020	177km ± 117	East-SE (122° ± 90)	1% ± 17	18% ± 34
Coastal (8 species)	2014	327km ± 81	NW-North (335° ± 10)	-49% ± 28	-26% ± 32
	2015	427km ± 133	NW-North (337° ± 5)	-59% ± 35	-35% ± 40
	2019	148km ± 166	South-SW (195° ± 101)	44% ± 141	2% ± 52
	2020	170km ± 228	West-NW (294° ± 77)	58% ± 174	-9% ± 72
Southern (2 species)	2014	183km ± 135	NW-North (343° ± 50)	-18% ± 5	-27% ± 23
	2015	347km ± 220	North (360° ± 13)	-32% ± 7	-48% ± 13
	2019	392km ± 30	NE-East (65° ± 8)	-38% ± 5	-43% ± 12
	2020	209km ± 34	NE-East (84° ± 30)	-14% ± 19	-25% ± 2

Reviewer 1, Comment 37.

-line 349: The Authors provided information here and in the Reporting Summary that R code is available through the GitHub repository, but it was not available during the review (404-page not available). It would help understand some of the steps of the analysis, however, the study doesn’t introduce novel algorithms or methods, thus R code review was not crucial.

Response Reviewer 1, Comment 37. Our apologies for this oversight. We have changed the repository from private to public https://github.com/HeatherWelch/MHW_impacts_top_predators

References:

Barker, N. K. S., Cumming, S. G., and Darveau, M. 2014. Models to predict the distribution and abundance of breeding ducks in Canada. *Avian Conservation and Ecology*, 9.

Barry, S., and Elith, J. 2006. Error and uncertainty in habitat models. *Journal of Applied Ecology*, 43: 413–423.

Woodman, S. M., Forney, K. A., Becker, E. A., DeAngelis, M. L., Hazen, E. L., Palacios, D. M., and Redfern, J. V. 2019. esdm: A tool for creating and exploring ensembles of predictions from species distribution and abundance models. *Methods in Ecology and Evolution*, 10: 1923–1933.

Reviewer #2 (Remarks to the Author):

Reviewer 2, Comment 1.

Given the increasing occurrence and prevalence of marine heatwaves - MHWs - it has been surprising that their impact on the distribution of marine animals has received so little attention. This is a very timely submission, that I believe will be of significance to the field, especially given that it is the first look at how the responses of marine megafauna might vary in response to short-term extreme events.

The results are, hence, quite novel, with the authors predicting distribution shifts during MHW years. The predicted impacts were, unsurprisingly, variable (range expansions/contractions) for the different species considered; but importantly, different MHWs were predicted to have impacted certain species (e.g., shearwaters; sea lions) differently.

It would have been ideal if tracking data spatio-temporally overlapped the MHWs, but the authors built a robust predictive framework to work around the gap in data. The authors did a great job exploring and presenting their results, especially when linking redistribution patterns in relation to political boundaries.

The manuscript is very well written and structured. I have no major comments and am supportive of its publication.

Response Reviewer 2, Comment 1. Thank you very much for your encouraging remarks.

MINOR COMMENTS:

Reviewer 2, Comment 2.

#1 I understand it is quite hard to make figures fit the limited space that journals provide, but Figure 1 should be made bigger since it is especially hard to see the kernel densities. The large number of species outlines for the coastal plot also does not help. This figure should also include a legend for the different species (like Figure 2).

Response Reviewer 2, Comment 2. We've done our best to increase the size of this figure by removing white space, and moving the map labels onto land to free up more space for the maps. We've also removed the species outlines, as this comment and Reviewer 2, Comment 7 noted they obscured the kernel densities:

Reviewer 2, Comment 3.

#2 The insets in Figure 3 should indicate what the axis refer to (percent change in area).

Response Reviewer 2, Comment 3. Done.

Reviewer 2, Comment 4.

#3 L192-194; In the first paragraph of the Discussion the summary of the results is sometimes vague. For example "[...] some predators were predicted to experience near total loss of habitat and range compression, while others were predicted to experience a two-fold habitat increase and significant range expansion." It would be better if authors mention which predators were predicted to experience range expansions/contractions.

Response Reviewer 2, Comment 4. We have edited this sentence to read:

“Predicted responses were highly variable across species and MHW (Fig. 2): some predators were predicted to experience near-total loss of habitat and range compression, e.g. bluefin tuna during the 2015 event, while others were predicted to experience a two-fold habitat increase and significant range expansion, e.g. California sea lion during the 2019 event.”

Reviewer 2, Comment 5.

#4 L209. "Sooty shearwaters were predicted to expand their range during the 2019 and 2020 events, likely associated with elevated primary productivity in coastal Alaskan waters". It is very minor comment, but Alaska is not shown in Figure 4 under the US EEZ.

Response Reviewer 2, Comment 5: True, we consolidated US west coast, Alaskan, and Hawaiian waters into one group to simplify Figure 4. We have added a new supplementary figure to support this statement:

“Sooty shearwaters were predicted to expand their range during the 2019 and 2020 events, likely associated with elevated primary productivity in coastal Alaskan waters (Fig. S1).”

A. Sea surface temperature

B. Oxygen at 200m

C. Mean primary productivity in the upper 200m

D. Log chlorophyll-a

Figure S1. Anomalies of sea surface temperature ($^{\circ}\text{C}$; A), oxygen concentration at 200m (mmol m^{-3} ; B), mean primary productivity in the upper 200m (mg m^{-3} ; C), and log chlorophyll-a (mg m^{-3} ; D) during each marine heatwave event. Anomalies were calculated from August-October in each year relative to a 2000-2020 August-October baseline.

I can only congratulate the authors on an excellent manuscript. Also, the supplementary materials are quite thorough, and the authors clearly aware of the caveats of the approach they used.

Reviewer #3 (Remarks to the Author):

This study investigates the effects on top predator species of multiple large-scale marine heatwaves, associated with ‘The Blob’, in the Northeast Pacific from 2014-2019. Telemetry data is used together with selected environmental variables to fit a species distribution model to predict distributions during marine heatwave events. The authors present different responses amongst species but also amongst heatwave events and potential consequences for ecosystem management due to shifting distributions into or out of US waters. The results highlight the urgent need to develop operational ocean management tools.

The manuscript is well written and results presented in a clear way. For full disclosure, this review focuses on the oceanographic aspects and marine heatwave definition of the manuscript. I recommend the manuscript for publication after some (potentially minor) revisions. Detailed comments are outlined Below.

General response reviewer 3. Thank you for reviewing the oceanography in our manuscript and for your positive and constructive feedback. We have worked to bring more of a regional oceanographic lens to our results.

Comments

I have a few more general comments about the definitions of marine heatwaves and associated terminology.

Reviewer 3, Comment 1.

The authors do not specifically detect MHWs here but build on existing literature that has identified the 4 big MHW events in the specific years that are referred to here. For completion it might be helpful to add some MHW metrics about those 4 events in the supplementary, e.g. start, end and peak date since marine heatwaves are defined as discrete events. I understand that these might not be as relevant for this study as the focus is on the months with the warmest total temperatures but if the term MHW is used it could be helpful to state them specifically or add a couple of sentences along the lines of what I mentioned above.

Response Reviewer 3, Comment 1. This is a good point. We have now edited this discussion paragraph to include start, end, and peak aerial extent data for each MHW:

“Prominent marine heatwaves (MHWs) in the North Pacific occurred in 2014, 2015, 2019, and 2020 (Extended Data Figure 1). Warm water conditions persisted across 2014 and 2015, however we elected to examine these years as distinct events due to their distinct warming patterns. Strongly positive temperature anomalies in the Gulf of Alaska appeared in October 2013, expanding to reach the US west coast in September 2014, penetrating to over 100m in depth across large portions of the region¹² (Fig. 1A). In fall of 2015, this pre-existing heatwave was amplified by a strong El Niño event, extending the spatial extent of the MHW westward towards Hawaii, increasing the depth penetration of warm anomalies, and producing the warmest tropical SST anomalies on record^{13,14}. This MHW dissipated in late 2015. The 2019 and 2020 MHWs originated in the Gulf of Alaska during late spring (April-May) and reached the US west coast in September, though there were differences in their seasonality and drivers¹⁵. The 2020 event was the second largest MHW on record (following the 2014 event), and at its maximum extent, the 2019 event was only 6% smaller than the 2014 event¹⁶. Despite their comparable sizes, the depth penetration of warm anomalies was much shallower in 2019 and 2020, reaching maximum depths of 40-50m¹². Both of these events rapidly retracted from the US west coast during October and November of their respective years, and dissipated below thresholds for marine heatwave classification¹¹ during December and January. Maximum aerial extents for each event were reached in September 2014, July 2015, September 2019, and September 2020.”

Reviewer 3, Comment 2.

Furthermore, the authors speak a few times about averages across each MHW event (e.g. in L299) which can be misleading. Instead, I would say something like Aug-Oct average during each MHW year. The years 2000-2020 are used as baseline to derive anomalies and the authors refer to it as climatology. I would not call it climatology as that term typically refers to a 30-year average, in particular for MHW studies.

Response Reviewer 3, Comment 2. Thank you, we have clarified that averages are calculated from Aug-Oct in each MHW year throughout the text, and replaced all instances of “climatology” with “baseline”.

Reviewer 3, Comment 3.

Furthermore, the authors justify the use of 2000-2020 with the limitation of the dataset, in particular because of the chlorophyll data, however, SST is available earlier and one could use a more common baseline (e.g. 1982-2010) to derive the anomalies to be more consistent with previous MHW work. Of course, it is under current debate how to best define a MHW (<https://pubmed.ncbi.nlm.nih.gov/37012469/>) and species’ responses will be very different depending on the thermal tolerance and potentially depend stronger on the absolute temperatures instead of anomalies? So, one overall question is, how sensitive are the results to the magnitude of anomalies and it was not clear to me if the habitat model was fit with SST anomalies or absolute SST values?

Response Reviewer 3, Comment 3. The habitat models were fit with absolute SST values as opposed to SST anomalies, thus none of our biological response metrics are sensitive to baseline length. While the available SST time-series is longer than that of the other variables, we have elected to maintain all anomaly calculations (e.g. Extended Data Figure 4) across the same time-series to allow anomalies to be comparable across variables and with biological responses (e.g. Fig 2).

Extended Data Figure 4. Anomalies of key dynamic variables during the four marine heatwave events. Anomalies of Sea surface temperature ($^{\circ}\text{C}$; A), log chlorophyll-a (mg m^{-3} ; B), oxygen at 200m (mmol m^{-3} ; C), and mean primary productivity in the upper 200m (mg m^{-3} ; D) measured over the 75th percentile kernels of each species. Anomalies were calculated from August-October in each year relative to a 2000-2020 August-October baseline.

Figure 2. Predicted impacts on top predator habitat within (columns, e.g. 2014) and among (rows, e.g. White shark) marine heatwave events measured using four metrics: A. displacement distance (kilometers), B. displacement direction (degrees, where 0/360 is north (N), 90 is east (E), 180 is south (S), and 270 is west (W)), C. range compression or expansion (percent change relative to baseline conditions), D. habitat area gain or loss (percent change relative to baseline conditions). All metrics were calculated from August-October in each MHW year relative to baseline conditions (August-October 2000-2020), see Table S2 for an analysis of metric uncertainty. Northern, Coastal, and Southern regional groupings indicate the geographies where the majority of the species telemetry data occurs.

Reviewer 3, Comment 4.

The authors make the point several times in the text that species impacts will vary across MHW events. It is worth highlighting in the discussion (e.g. L199-230) that MHWs are just a statistical construct, but they can be caused by very different physical mechanisms, which likely is reflected in other variables too. In the supplementary the authors mention that for example the events in 2019 and 2020 were much shallower than 2014 and 2015, which likely drives different responses. In my opinion it is crucial to discuss the different regional ocean processes associated with different types of marine heatwaves and how these can for example impact oxygen at depth though for example variability in the coastal upwelling. I am aware that a detailed discussion of the physical processes is likely out of scope for this study but still think that it would be important to have a short paragraph in the discussion dedicated to this.

Response Reviewer 3, Comment 4. We have added some regional oceanographic context to explain the patterns seen in sooty shearwaters and elephant seals:

“However, temperature anomalies were fairly consistent across the 75th kernels of sooty shearwaters and elephant seals during the four MHW events, yet each species experienced markedly different predicted impacts among MHWs. Sooty shearwaters were predicted to expand their range during the 2019 and 2020 events, likely associated with elevated primary productivity in coastal Alaskan waters. Elephant seals were predicted to experience large habitat loss during the 2014 and 2015 events, likely due to a reduction of primary productivity and subsurface oxygen in the north-central Pacific. The deep warm water anomalies of the 2014 and 2015 events decreased the solubility of oxygen and led to a near-surface decline in oxygen concentration⁷. The 2019 and 2020 events, in contrast, had much shallower temperature anomalies, and likely had a weaker impact on local biogeochemical signals and productivity. In general, variations in winds, air-sea gas exchange, circulation patterns and water column stratification can lead to very different physical and biogeochemical conditions among MHW events. The idiosyncratic responses of species to MHWs reflect these variable conditions and differing species-environment relationships (Extended Data Figure 5).”

And an additional supplementary figure showing conditions of oxygen, primary productivity, and chlorophyll-a during the MHW years:

A. Sea surface temperature

B. Oxygen at 200m

C. Mean primary productivity in the upper 200m

D. Log chlorophyll-a

Figure S1. Anomalies of sea surface temperature ($^{\circ}\text{C}$; A), oxygen concentration at 200m (mmol m^{-3} ; B), mean primary productivity in the upper 200m (mg m^{-3} ; C), and log chlorophyll-a (mg m^{-3} ; D) during each marine heatwave event. Anomalies were calculated from August-October in each year relative to a 2000-2020 August-October baseline.

We are in full agreement that variables beyond temperature are driving species response to MHWs, and have a paragraph devoted to this in the discussion:

“MHW-driven species redistributions are most commonly ascribed to the relocation of preferred temperature conditions¹⁷⁻¹⁹. However, our findings suggest that a multi-variable approach allows for

additional ecological inferences with regard to MHW biodiversity impacts. Many of the predators examined here have broad thermal tolerances, and are distributed in warmer waters elsewhere in the Pacific than those encountered during the MHWs²⁰. Thus it is likely that anomalous conditions of variables beyond temperature (Extended Data Figure 4) are driving their responses. Indeed, a comparable suite of temperature-only models had significantly worse predictive performance on novel validation data than the multivariate models (median Area Under the Receiver Operator Characteristic Curve of 0.6 in the temperature-only models vs 0.8 in the multivariate models, t-test p-value <0.05). In following with evidence that temperature alone cannot account for species responses to climate change^{2,21}, we suggest that a multi-variable approach is critical to capture species responses to short-term warming. Several programs exist to monitor and forecast extreme ocean warming based on observed and predicted ocean temperatures^{10,22}, and these results indicate the utility of concurrently tracking ocean conditions such as oxygen and productivity for a more nuanced understanding of possible species impacts.”

Line-based comments

Reviewer 3, Comment 5.

L135ff: Before talking about the effects of MHWs, I think it would be useful to briefly mention that this study focuses on Aug-Oct only in each MHW year. I am aware that the specifics of each MHW are not the focus of the study but I believe it will help the reader with the interpretation of the results.

Response Reviewer 3, Comment 5. Thank you, we have indicated that metrics are calculated from Aug-Oct in the caption of Fig. 2, which displays the metrics discussed in this section:

“All metrics were calculated from August-October in each MHW year relative to baseline conditions (2000-2020).”

Figures

Reviewer 3, Comment 6.

Fig1: This might be personal taste, but I believe maps should always include some meridians and parallels for reference. The authors are likely very familiar with the geography of the region but not all readers will be.

Response Reviewer 3 Comment 6. This is a good idea, we have made this change.

Reviewer 3, Comment 7.

Fig3: Add axis labels or change tick labels to include °N etc.

Response Reviewer 3, Comment 7. Done:

Reviewer 3, Comment 8.

Figure S1: The panels are very small and have low resolutions which makes it impossible to read the numbers on the colorbar.

Response Reviewer 2, Comment 1. Thank you - we have increased the size of this figure and improved the resolution

A. Telemetry data used in model fitting

B. Independent data used in model validation

References

1. Pinsky, M., Worm, B., Fogarty, M., Sarmiento, J. & Levin, S. Marine Taxa Track Local Climate Velocities. *Science* **341**, 1239–42 (2013).
2. McHenry, J., Welch, H., Lester, S. E. & Saba, V. Projecting marine species range shifts from only temperature can mask climate vulnerability. *Glob. Change Biol.* **25**, 4208–4221 (2019).
3. Morley, J. W. *et al.* Projecting shifts in thermal habitat for 686 species on the North American continental shelf. *PLOS ONE* **13**, e0196127 (2018).
4. Abrahms, B. *et al.* Dynamic ensemble models to predict distributions and anthropogenic risk exposure for highly mobile species. *Divers. Distrib.* **25**, 1182–1193 (2019).
5. Hazen, E. L. *et al.* A dynamic ocean management tool to reduce bycatch and support sustainable fisheries. *Sci. Adv.* **4**, eaar3001 (2018).
6. Becker, E. A. *et al.* Predicting cetacean abundance and distribution in a changing climate. *Divers. Distrib.* **25**, 626–643 (2019).
7. Mogen, S. C. *et al.* Ocean Biogeochemical Signatures of the North Pacific Blob. *Geophys. Res. Lett.* **49**, e2021GL096938 (2022).
8. Elith, J., Leathwick, J. R. & Hastie, T. A working guide to boosted regression trees. *J. Anim. Ecol.* **77**, 802–813 (2008).
9. Jacox, M. G., Alexander, M. A., Bograd, S. J. & Scott, J. D. Thermal displacement by marine heatwaves. *Nature* **584**, 82–86 (2020).
10. Jacox, M. G. *et al.* Global seasonal forecasts of marine heatwaves. *Nature* **604**, 486–490 (2022).
11. Hobday, A. J. *et al.* A hierarchical approach to defining marine heatwaves. *Prog. Oceanogr.* **141**, 227–238 (2016).
12. Scannell, H. A., Johnson, G. C., Thompson, L., Lyman, J. M. & Riser, S. C. Subsurface Evolution and Persistence of Marine Heatwaves in the Northeast Pacific. *Geophys. Res. Lett.* **47**, e2020GL090548 (2020).
13. Jacox, M. G. *et al.* Impacts of the 2015–2016 El Niño on the California Current System: Early assessment and comparison to past events. *Geophys. Res. Lett.* **43**, 7072–7080 (2016).
14. Rudnick, D. L., Zaba, K. D., Todd, R. E. & Davis, R. E. A climatology of the California Current

- System from a network of underwater gliders. *Prog. Oceanogr.* **154**, 64–106 (2017).
15. Amaya, D. J., Miller, A. J., Xie, S.-P. & Kosaka, Y. Physical drivers of the summer 2019 North Pacific marine heatwave. *Nat. Commun.* **11**, 1903 (2020).
 16. Weber, E. D. *et al.* State of the California Current 2019–2020: Back to the Future With Marine Heatwaves? *Front. Mar. Sci.* **8**, 1081 (2021).
 17. Tanaka, K. R. *et al.* North Pacific warming shifts the juvenile range of a marine apex predator. *Sci. Rep.* **11**, 3373 (2021).
 18. Carroll, G. *et al.* Flexible use of a dynamic energy landscape buffers a marine predator against extreme climate variability. *Proc. R. Soc. B Biol. Sci.* **288**, 20210671 (2021).
 19. Welch, H. *et al.* Environmental indicators to reduce loggerhead turtle bycatch offshore of Southern California. *Ecol. Indic.* **98**, 657–664 (2019).
 20. Block, B. A. *et al.* Tracking apex marine predator movements in a dynamic ocean. *Nature* **475**, 86–90 (2011).
 21. Niella, Y., Butcher, P., Holmes, B., Barnett, A. & Harcourt, R. Forecasting intraspecific changes in distribution of a wide-ranging marine predator under climate change. *Oecologia* **198**, 111–124 (2022).
 22. Liu, G., Strong, A., Skirving, W. & Arzayus, F. Overview of NOAA coral reef watch program's near-real time satellite global coral bleaching monitoring activities. *Proc 10th Int Coral Reef Symp* **1**, 1783–1793 (2005).

REVIEWERS' COMMENTS

Reviewer #1 (Remarks to the Author):

I re-evaluated the manuscript after the first round of revisions. The authors did an excellent job addressing the reviewers' comments and suggestions. I appreciate the effort put to conduct additional tests, including sensitivity runs. In a few cases, when authors did not follow my suggestions, they provided very reasonable rebuttals, which I agree with. Added or modified parts of the text improved the clarity and provided interesting perspectives to the discussion. The parallel to the hurricanes seems very useful and helps the reader understand the issue. The quality of the figures has been improved. Overall, after the last changes, the manuscript gained a lot of strength. I am satisfied with all the changes made.

My only suggestion would be to consider mentioning the general results of the new analyses presented in the supplementary materials in the main body of the text. For example, it is reported in the supplementary materials that "Results indicate that species explains more of the variance than MHW for percent change range, percent change area, and distance habitat metrics", or "Ratios showed high alignment between daily observed and predicted presences (mean 1.5 +/- 1.3 SD), indicating strong capacity to generate predictions in a near-real time framework", but these results are not mentioned in the main body of text. What is the general interpretation of the results of the sensitivity test? You may consider including very brief information on each result in appropriate sections with reference to the supplementary materials.

I don't have any more comments. If authors find my suggestion useful, very minor modifications can be done to integrate summaries of the new results. Otherwise, I believe this manuscript can be considered for publication in the journal.

Reviewer #3 (Remarks to the Author):

The authors have addressed all my comments – I recommend the manuscript for publication as is. I also want to commend and thank the authors for the thorough response to all reviewers; it was also very helpful to have the changed/added text and figures within the document.

This study is an important contribution as to the impacts of marine heatwaves on top predators and highlights the need for effective multi-species ocean management as well as multi-variable analyses and the consideration of the depth-extent for marine heatwave analyses.

I again commend the authors for a very well written manuscript and clear figures to support the presented results.

REVIEWERS' COMMENTS

Reviewer #1 (Remarks to the Author):

I re-evaluated the manuscript after the first round of revisions. The authors did an excellent job addressing the reviewers' comments and suggestions. I appreciate the effort put to conduct additional tests, including sensitivity runs. In a few cases, when authors did not follow my suggestions, they provided very reasonable rebuttals, which I agree with. Added or modified parts of the text improved the clarity and provided interesting perspectives to the discussion. The parallel to the hurricanes seems very useful and helps the reader understand the issue. The quality of the figures has been improved. Overall, after the last changes, the manuscript gained a lot of strength. I am satisfied with all the changes made.

My only suggestion would be to consider mentioning the general results of the new analyses presented in the supplementary materials in the main body of the text. For example, it is reported in the supplementary materials that “Results indicate that species explains more of the variance than MHW for percent change range, percent change area, and distance habitat metrics”, or “Ratios showed high alignment between daily observed and predicted presences (mean 1.5 +/- 1.3 SD), indicating strong capacity to generate predictions in a near-real time framework”, but these results are not mentioned in the main body of text. What is the general interpretation of the results of the sensitivity test? You may consider including very brief information on each result in appropriate sections with reference to the supplementary materials. I don't have any more comments. If authors find my suggestion useful, very minor modifications can be done to integrate summaries of the new results. Otherwise, I believe this manuscript can be considered for publication in the journal.

Response Reviewer #1. Thank you for your encouragement, and for improving the quality and clarity of our manuscript during two rounds of review. In following with the above suggestion, we now reference our analysis of daily predictive capacity in the discussion:

“Our results indicate that species responses to MHWs are highly variable yet predictable: our models performed well through extensive validation across space, time, and on novel data ((Supplementary Figs. 6,7; Supplementary Table 3)”.

We have elected to not report results from the sensitivity analysis to data used in model fitting, but these supplementary tables are referenced in the relevant figures (Fig. 2 and Fig. 4). We have also elected to not include the analysis of variance between vs among marine heatwaves in the main text, as we were not able to complete this analysis for direction. We feel there is value in maintaining a simple and straightforward narrative in the main text to increase accessibility for a broad readership; while including more detailed and nuanced analyses/results in the supplements for subject experts.

Reviewer #3 (Remarks to the Author):

The authors have addressed all my comments – I recommend the manuscript for publication as is. I also want to commend and thank the authors for the thorough response to all reviewers; it was also very helpful to have the changed/added text and figures within the document.

This study is an important contribution as to the impacts of marine heatwaves on top predators and highlights the need for effective multi-species ocean management as well as multi-variable analyses and the consideration of the depth-extent for marine heatwave analyses.

I again commend the authors for a very well written manuscript and clear figures to support the presented results.

Response Reviewer #3. We appreciate your help improving the clarity and messaging of our manuscript.